# Crystal and electronic facet analysis of ultrafine Ni$_2$P particles by solid-state NMR nanocrystallography

Wassilios Papawassiliou [1], José P. Carvalho [1], Nikolaos Panopoulos[2], Yasser Al Wahedi [3✉], Vijay Kumar Shankarayya Wadi [3], Xinnan Lu [3], Kyriaki Polychronopoulou[4,5], Jin Bae Lee[6], Sanggil Lee[6], Chang Yeon Kim[6], Hae Jin Kim [6], Marios Katsiotis [3], Vasileios Tzitzios [2], Marina Karagianni [2], Michael Fardis[2], Georgios Papavassiliou [2✉] & Andrew J. Pell [1,7✉]

Structural and morphological control of crystalline nanoparticles is crucial in the field of heterogeneous catalysis and the development of "reaction specific" catalysts. To achieve this, colloidal chemistry methods are combined with ab initio calculations in order to define the reaction parameters, which drive chemical reactions to the desired crystal nucleation and growth path. Key in this procedure is the experimental verification of the predicted crystal facets and their corresponding electronic structure, which in case of nanostructured materials becomes extremely difficult. Here, by employing $^{31}$P solid-state nuclear magnetic resonance aided by advanced density functional theory calculations to obtain and assign the Knight shifts, we succeed in determining the crystal and electronic structure of the terminating surfaces of ultrafine Ni$_2$P nanoparticles at atomic scale resolution. Our work highlights the potential of ssNMR nanocrystallography as a unique tool in the emerging field of facet-engineered nanocatalysts.

---

[1] Department of Materials and Environmental Chemistry, Arrhenius Laboratory, Stockholm University, Stockholm, Sweden. [2] Institute of Nanoscience and Nanotechnology, National Center for Scientific Research "Demokritos", Attiki, Greece. [3] Department of Chemical Engineering, Khalifa University, Abu Dhabi, United Arab Emirates. [4] Center for Catalysis and Separations (CeCaS), Khalifa University, Abu Dhabi, United Arab Emirates. [5] Department of Mechanical Engineering, Khalifa University, Abu Dhabi, United Arab Emirates. [6] Electron Microscopy Research Center, Korea Basic Science Institute, Yuseong-gu, Daejeon, Republic of Korea. [7] Centre de RMN à Très Hauts Champs de Lyon (UMR 5280 CNRS/ENS Lyon/Université Claude Bernard Lyon 1), Université de Lyon, Villeurbanne, France. ✉email: yasser.alwahedi@ku.ac.ae; g.papavassiliou@inn.demokritos.gr; andrew.pell@mmk.su.se

Shape and size engineering of functional nanocrystalline materials is an area of major scientific and technological interest[1–3]. This is because in many important applications which rely on surface structure and chemistry, such as heterogeneous catalysis, gas sensing, and energy conversion and storage, the properties of the materials can be tailored by controlling the size, crystal structure, and morphology of the external surfaces of the constituent particles[4–6]. For heterogeneous catalysis, scaling down the particle size not only increases the number of catalytic sites, but also modifies the electronic properties. Furthermore, the catalytic reactivity and selectivity changes by modifying the arrangement and coordination of the surface atoms, thus becoming very sensitive to the enclosing crystal facets. In this context a lot of effort has been devoted in order to understand the role of the facet morphology and crystal structure on the catalytic reactivity of $Ni_2P$ nanoparticles, a staple in the area of research of many important catalytic processes, such as the electrocatalytic hydrogen evolution reaction (HER)[7] and oxygen evolution reaction (OER)[8], the water-gas shift reaction[9], as well as the hydrodeoxygenation and the hydrodesulfurization (HDS) of hydrocarbons[10].

$Ni_2P$ adopts an ABAB two-layer arrangement along the [0001] direction of the crystal structure, as shown in Fig. 1d, in which the alternating layers have stoichiometry $Ni_3P_2$ and $Ni_3P$, competing for the possession of the (0001) surface. First principles calculations[11] have shown that the $Ni_3P_2$ termination is energetically favored in comparison to the $Ni_3P$ one; similarly, experiments on $Ni_2P$ single crystals[12,13] supported by DFT calculations[14], indicate that the hexagonal (0001) surface is terminated primarily by P-covered $Ni_3P_2$ ($Ni_3P$-P), with P adatoms stabilizing the surface via dangling bonds to threefold Ni atoms. Nevertheless, $Ni_3P$ facets were occasionally observed under strong high temperature treatment[15]. It is well established by now, that exposed (0001) facets contribute significantly to the higher catalytic activity of the $Ni_2P$ nanoparticles[7,9,16]. However, the stability and termination of the (0001) facet in terms of potential adatoms,

as well as of other terminating $Ni_2P$ surfaces depends on the experimental synthesis conditions, and this dependence is not yet sufficiently well understood. For instance, ab initio atomistic thermodynamic studies on the stability of different low-Miller-index $Ni_2P$ surfaces, by considering different temperatures, pressures, and chemical potentials during synthesis, have shown that the equilibrium morphology of $Ni_2P$ nanoparticles comprises of a variety of surfaces, predominantly with the (0001), (10$\bar{1}$0), and (101$\bar{1}$) facets. An important finding of this study is, that under an excess of phosphorous during the synthesis, nanoparticles are predicted to acquire hexagonal rod-like shapes, with frontal facets the (0001-A) and lateral facets the (10$\bar{1}$0)[17]. Similarly in the case of $Ni_2P$ nanowires, primarily the (0001), (10$\bar{1}$0), and (1$\bar{1}$20) surfaces are exposed[17,18].

As solid-state nuclear magnetic resonance (ssNMR) is sensitive to the structure and electronic environment at the atomic scale, it is able to determine the distinct surface facets in nanosized particles as well as distinguish between these and the bulk-like interior of the nanoparticle[19–22]. In the case of nanocrystalline metallic powders, the strength of ssNMR lies in its ability to probe the interactions which couple the conduction electrons with the nuclear spins, via the Knight shift[23] $K = K_{FC} + K_{dip} + K_{orb}$. In this formula, $K_{orb}$ is the orbital part of the Knight shift, induced by the response of the orbital motions of all electrons to the applied external magnetic field, and $K_{FC}$ and $K_{dip}$ are the Fermi-contact and spin-dipolar parts of the Knight shift, respectively, due to the contact and dipolar parts of the hyperfine interaction of the nuclear spins with the net electron magnetic moment of the conduction electrons, again induced by the external magnetic field[24,25].

In metallic systems, to first noninteracting electrons approximation, $K_{FC}$ is approximated by:

$$K_{FC} = \frac{8\pi}{3}\langle|\varphi_s(0)|^2\rangle_F \mu_B^2 N(E_F), \tag{1}$$

where $|\varphi_s(0)|^2$ is the probability density of the s–band electrons

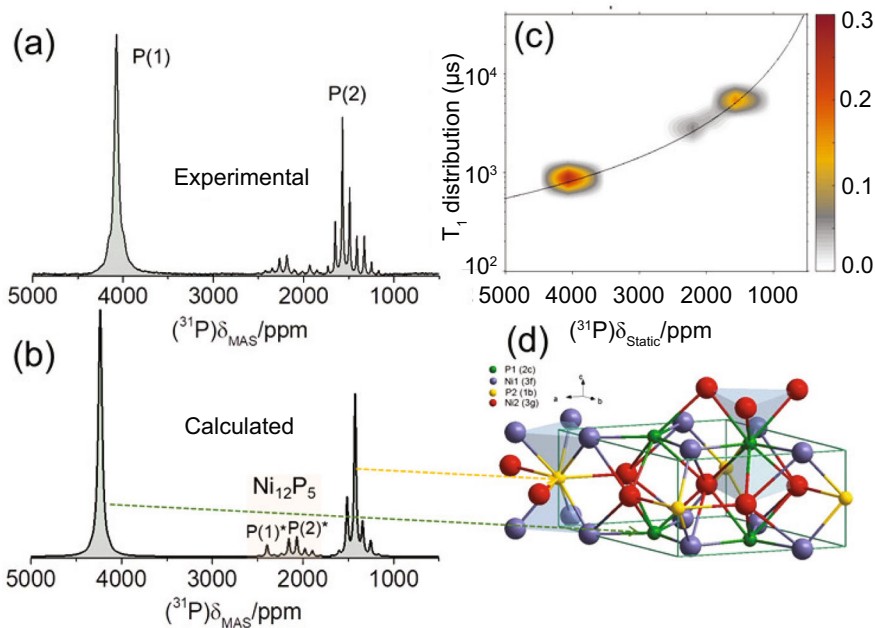

**Fig. 1 $^{31}$P ssNMR, spin–lattice relaxation analysis and DFT-calculated spectra of microcrystalline $Ni_2P$. a** $^{31}$P NMR spectrum of microcrystalline $Ni_2P$ at 14 kHz MAS. **b** Simulated $^{31}$P MAS NMR spectrum with the DFT-calculated Knight shifts. **c** Spin–lattice relaxation times $T_1$ vs. NMR frequency for microcrystalline $Ni_2P$ under static conditions. The superimposed curve is fitted according to the Korringa relation (Eq. S3 in Supplementary Note 5). **d** Visualization of the crystallographic unit cell and its nonequivalent Ni and P sites.

within the nuclear volume evaluated near the Fermi level, $N(E_F)$ is the s-electron projected density of states (pDOS) at the Fermi level, and $\mu_B$ is the Bohr magneton. This Fermi-contact term is often the dominant contribution to the Knight shift in metallic materials[24,26–28] and equips NMR with the opportunity to experimentally probe the local electronic density of states near the Fermi level, with obvious significance for chemistry and catalysis, as the ratio between the density of localized and delocalized electrons near the Fermi level greatly affects the catalytic properties[29].

In this context, we have employed density-functional-theory (DFT)-assisted $^{31}$P ssNMR combined with experimental and theoretical transmission electron microscopy (TEM) (i.e., experimental and DFT-calculated TEM) to (i) identify the dominant crystal facets of ultrafine $Ni_2P$ nanoparticles synthesized in phosphorous excess conditions, which is predicted to grow nanocrystals with the simplest morphology[17], and (ii) visualize the subtle structural and electronic changes that take place on the exposed surfaces, with respect to the bulk interior of the nanoparticles. Results show that the $Ni_3P_2$–terminated surfaces, from now on labeled as (0001-A) crystal facets, prevail over the $Ni_3P$-terminated surfaces, labeled as (0001-B). Most importantly, the nanoparticles appear to have a thin platelet-like shape with frontal (0001-A) and lateral (10$\bar{1}$0) surfaces in ultrasmall sizes, whereas by increasing size, they grow preferentially along the [0001] direction, exposing predominately the (10$\bar{1}$0) facets, in agreement with theoretical predictions[17]. Our NMR-confirmed DFT calculations show that on both terminating surfaces, the Ni d-electron bands shift towards the Fermi level, indicating that the reported enhanced catalytic activity of the $Ni_2P$ nanoparticles[7–10,16] to great extent originates from the changes in the electronic properties of the surface.

This is the first time that facet analysis of a transition metal nanosized catalyst and the relevant electronic changes are experimentally verified, showcasing that ssNMR nanocrystallography is an emerging tool in the study of metallic nanocatalysts.

## Results

### $^{31}$P ssNMR and DFT calculations of microcrystalline (bulk) $Ni_2P$.

The potential of DFT-assisted ssNMR to acquire accurate structural and electronic information on transition metal phosphides is showcased in Fig. 1. Microcrystalline $Ni_2P$ is known to adopt the hexagonal structure assigned to the $Fe_2P$ structure type, with space group P-62m[30]. The structure can be visualized as an alternation of two nonequivalent atomic layers along the [0001] direction with stoichiometry of $Ni_3P_2$ (0001-A) and $Ni_3P$ (0001-B). Nickel and phosphorous atoms in the two layers are labeled as Ni(1)/P(1) (A-layer) and Ni(2)/P(2) (B-layer), respectively (Fig. 1d). Figure 1a presents the $^{31}$P magic-angle-spinning (MAS) ssNMR spectrum of microcrystalline $Ni_2P$, and comprises two primary resonances: one with an isotropic shift at 4072 ppm, which is attributed to the P(1) site, and the second at 1490 ppm, attributed to P(2), in agreement with previous studies[31]. In addition, two weak NMR resonances accompanied by their respective spinning-sideband (SSB) manifolds are observed between 1800 and 2250 ppm and are attributed to the two nonequivalent P sites of a minority $Ni_{12}P_5$ phase[31]. To confirm this assignment, the NMR Knight shifts were calculated by DFT, using the $Ni_2P$ atomic positions determined from the Rietveld analysis of the XRD pattern (Supplementary Fig. 1a). The resulting simulated $^{31}$P NMR spectrum (Fig. 1b) strongly resembles the experimental $^{31}$P NMR spectrum.

The significant difference in the Knight shift $K$ of the P(1) and P(2) sites is caused by the different Ni–P bond lengths (5.85 Å for Ni(2)-P(2) and 3.38 Å for Ni(1)–P(1)), leading to higher projected DOS (pDOS) of the s orbitals of the P(1) atoms at

the Fermi level $N(E_F(|s>))$, than in the P(2) atoms, as shown in Supplementary Fig. 6c, d. Notably, the ratio of pDOS(P(1))/pDOS (P(2)) $\approx 2.67$ of the s-electrons is approximately equal to the ratio of experimental Knight shifts $K(P(1))/K(P(2)) \approx 2.8$, indicating that conduction electrons, especially of P(1) atoms, appear in excess on the $Ni_3P_2$ A-layer in comparison to the P(2) atoms on the $Ni_3P$ B-layer. Furthermore, the experimental Knight shift values are in agreement with the calculated total (isotropic) $^{31}$P NMR Knight shift, referenced according to the relation $K_{calc} = \sigma_{ref} - \sigma_{calc}$. In this formula $\sigma_{ref} = 220.23$ ppm is the reference $^{31}$P NMR shielding, calculated with the procedure used by Mayo et al.[32] (see Supplementary Fig. 11), and $\sigma_{calc}$ the total DFT-calculated $^{31}$P NMR shielding in $Ni_2P$, which is equal to $\sigma_{calc} = \sigma_{FC} + \sigma_{dip} + \sigma_{orb}$[23]. According to the DFT calculations, the Fermi-Contact shielding $\sigma_{FC}$ is the primary contribution responsible for the large NMR shift difference between the two nonequivalent sites ($\sigma_{FC}$ is equal to −4076 ppm for P(1) and −1063 ppm for P (2)), while the orbital contribution is smaller ($\sigma_{orb} = 62.45$ ppm for P(1) and −146.78 for P(2)). The dipolar term $\sigma_{dip}$ is negligible, of the order of a few ppm for both P(1) and P(2). At the same time, the P(2) atoms on the $Ni_3P$ layer probe an increased orbital current shift anisotropy ($\Delta\delta = -242.41$ ppm) compared to P(1) ($\Delta\delta = -39.35$ ppm), as manifested by the broader SSB manifold of this resonance (anisotropies are given in the Haeberlen convention[33]). Detailed breakdown of the experimental and calculated parameter values is provided in Supplementary Table 1. In addition, the $^{31}$P NMR signals from the minority $Ni_{12}P_5$[34] phase were reproduced by DFT calculations in a similar way. In this case the two sites P(1)* and P(2)*, which appear in ratio 1:4, acquire $\sigma_{FC}$ values −2110 and −1665 ppm, respectively, $\sigma_{orb}$ −75 and −186 ppm, whereas shift anisotropies $\Delta\delta$ are calculated as 139 and −344 ppm.

The $^{31}$P NMR Knight shift analysis is in full accordance with the spin–lattice relaxation time $T_1$ measurements performed in the temperature range 290 to 500 K, which are presented in Fig. 1c and Supplementary Fig. 17; these data demonstrate excellently the metallic character of $Ni_2P$, since the Knight shifts and relaxation times together follow the Korringa relation[28], described in Supplementary Note 5. Specifically, $T_1(P(2))/T_1(P(1)) \approx \left(K(P(1))/K(P(2))\right)^2 \approx 7.85$, in agreement with the Korringa relation in Eq. (S3).

### Scaling down to nanosized particles.

On the basis of the results presented above, the question that arises is whether ssNMR is able to monitor the evolution of the crystal and electronic structure of $Ni_2P$, when the system dimensions shrink from a microcrystalline to a nanocrystalline structure, as this scaling down is predicted to expose a greater particle surface area, and a variety of crystal facets[17]. For this reason, four $Ni_2P$ nanoparticle systems were synthesized with particle size ranging from 4.3 to 40 nm and studied by combining DFT-assisted NMR and HRTEM. For the synthesis we used the co-precipitation method in the presence of phosphorous excess, which according to theoretical studies, drives the system in the simplest possible facet morphology with prevalent the (0001) and (10$\bar{1}$0) facets[17]. It is noticed that the presence of a $Ni_{12}P_5$ impurity phase that many times occurs as a byproduct during the synthesis of the nanoparticles is excluded, as clearly shown by comparing the XRD patterns of the $Ni_2P$ nanoparticles with the relevant XRD pattern of $Ni_{12}P_5$ nanoparticles, explicitly synthesized for this purpose (Supplementary Figs. 1b, 3). Details on the synthesis and characterization of the $Ni_2P$ nanoparticles are provided in the "Methods" section and the Supplementary Information.

In general, the ability of XRD for accurately and precisely determining the crystal structure of the $Ni_2P$ nanoparticles is

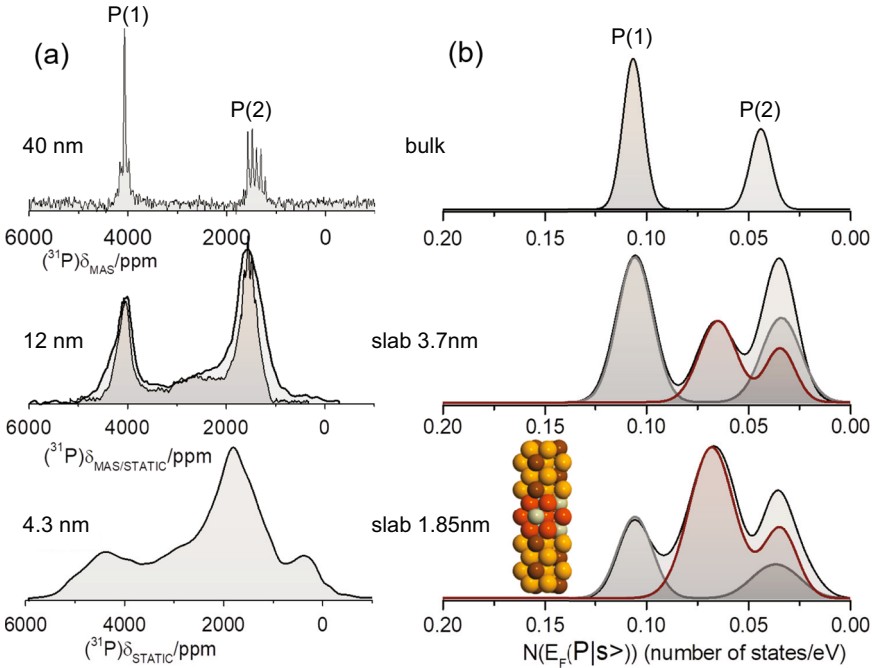

**Fig. 2 Correlating $^{31}$P ssNMR spectra with DFT calculations. a** From top to bottom: $^{31}$P MAS at 14 kHz MAS and static NMR spectra of nanocrystalline Ni$_2$P with mean diameters $d \approx 40$, 12, and 4.3 nm. In the mid panel the overlaid spectrum is the relevant MAS spectrum. **b** From top to bottom: visualization of the phosphorous s-electrons pDOS distribution at the Fermi level $N(E_F(|s>))$, in bulk Ni$_2$P and two slabs with thickness 3.7 and 1.85 nm, respectively, with Ni$_3$P$_2$ terminating layers (0001-A). Lines in red and gray colors are the relevant pDOS contributions from surface and core-bulk-like phosphorous atoms, respectively. The bottom right insert displays the (0001-A) super-cell with 50 atoms (1.85 nm), with the orange and brown spheres representing the surface nickel and phosphorus atoms, while red and gray spheres represent the bulk nickel and phosphorus atoms respectively.

limited upon reducing the particle size, because the XRD peaks are broadened and merge, according to the Scherrer relation, as demonstrated in Supplementary Fig. 1b. Fortunately, ssNMR still delivers important information, again in the form of Knight shifts, which are sensitive to the subtle structural and electronic changes induced by reducing the particle size, as demonstrated by the $^{31}$P NMR spectra in Fig. 2. While the NMR spectrum of Ni$_2$P nanoparticles with mean size of $d \approx 40$ nm is similar to that of microcrystalline Ni$_2$P (upper panel in Fig. 2a), upon reduction of the nanoparticle size to 12 nm, the NMR signals broaden, and an extended spectral feature appears between the P(1) and P(2) resonances. This feature grows at the expense of the P(1) resonance, which has an integral that is reduced in comparison to P(2), similarly to observations previously reported for ZnSe nanoparticles[35]. Notably, there is no information gain with MAS spinning, at least for a spinning frequency of 14 kHz, as seen in the mid panel of Fig. 2a. By further reducing the particle size the NMR signal becomes even broader extending in the spectral range between 0 and 5000 ppm, whilst most of the signal intensity of the P(1) resonance is shifted to lower frequency (lower panel of Fig. 2a).

To correlate the changes in the NMR lineshapes with alterations in the surface electron states, DFT calculations of the electron DOS were performed on a number of Ni$_2$P slabs with different exposed surfaces. DOS plots of the two most probable exposed surfaces, the (0001-A) with terminating Ni$_3$P$_2$ surfaces, and (10$\bar{1}$0) are presented in the Supplementary Figs. 8–10. Figure 2b displays the distribution of the pDOS of the s-electron states at the Fermi level $N(E_F(|s>))$ for all nonequivalent phosphorous atoms in three atomic arrangements: two (0001-A) terminating slabs with thickness 1.85 nm (super-cell of 50 atoms) and 3.71 nm (super-cell of 104 atoms), respectively, and 3D periodic (bulk) Ni$_2$P. A remarkable correlation of the pDOS

plots with the NMR lineshapes can be observed, especially in case of ultrasmall nanoparticles, which follows the relation between Knight shift and pDOS in Eq. (1). In particular, in the case of bulk Ni$_2$P two peaks are observed at pDOS values 0.108 states/eV and 0.043 states/eV, which are attributed to the P(1) and P(2) atomic sites, respectively. Notably, in the case of the two slabs, the pDOS of the P(1) surface states (red lines) appears to shift towards the P(2) DOS peak. Furthermore, by increasing the slab thickness the central "bulk"-like states (gray lines) gain in intensity, resembling the changes of the $^{31}$P NMR lineshapes with increasing size in Fig. 2a. Supplementary Figs. 7, 8 show clearly that the strong shift of the s-electrons pDOS of surface P(1) atoms is associated with substantial structural and electronic changes in the vicinity of the (0001-A) terminating surface: whilst the central atoms keep the same Wyckoff symmetry positions in the P-62m[30] space group (Ni(1)(3f), Ni(2)(3g), P(1)(2c), P(2)(1b)), the symmetry of both Ni atoms at the surface coalesces to Ni(6i), while the symmetry of the two phosphorus atoms changes to P(1)(4 h) and P(2)(2e). These crystal symmetry changes are reflected in the projected DOS of the surface atoms. Both Ni(1) and Ni(2) in the middle of the slab exhibit almost the same p-DOS as bulk Ni$_2$P, while surface Ni(1) atoms show a remarkable shift of their pDOS towards the Fermi level, acquiring a similar pDOS profile to the Ni(2) surface atoms. Accordingly, the pDOS of surface P(1) atoms is seen to shift towards the Fermi level.

The second important terminating surface is the (10$\bar{1}$0) surface. Figure 3a displays the phosphorous s-electrons pDOS distribution at the Fermi level $N(E_F(|s>))$ of a (10$\bar{1}$0) facet slab comprising a super-cell of 54 atoms. Similarly to the (0001-A) termination, the pDOS of the surface P(1) atoms shifts towards that of P(2), resulting in the formation of a characteristic doubly-peaked distribution with a strong peak close to the position of P(2). Notably, in both the (0001-A) and (10$\bar{1}$0) studied cases, the

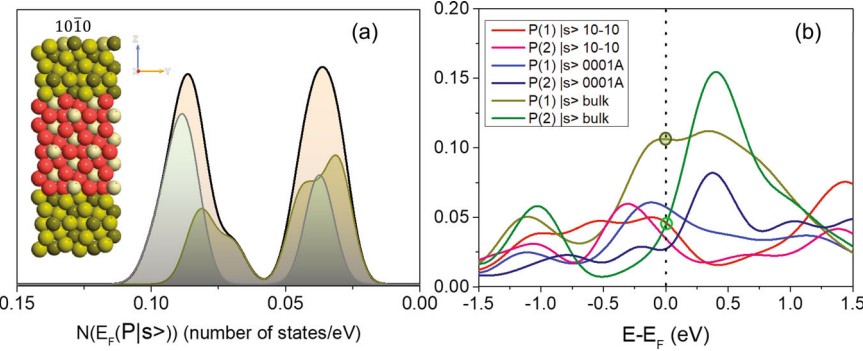

**Fig. 3 DFT calculated pDOS at the Fermi level. a** The phosphorous s-electrons pDOS distribution at the Fermi level $N(E_F(|s>))$ of the (10$\bar{1}$0) terminating slab. The inset shows the relevant 54-atoms super-cell, where the dark green and light green spheres represent the surface P and Ni respectively, while the red and gray spheres represent the bulk Ni and P. **b** The s-electrons pDOS of the P(1) and P(2) atomic positions in three different configurations: the (0001 A), (10$\bar{1}$0), and bulk Ni$_2$P. Circles indicate the pDOS values for bulk P(1) and P(2) atoms.

pDOS of the surface d-electrons shifts toward the relevant Ni(2) pDOS, while at the slab center the bulk Ni$_2$P pDOS is retained (Supplementary Figs. 9, 10).

Figure 3b compares the s-electron pDOS of the outermost (surface) P(1) and P(2) atoms for the (0001-A), and (10$\bar{1}$0) slabs, with that of the bulk Ni$_2$P. For reference, dark and light green circles indicate the pDOS at the Fermi level for bulk Ni$_2$P. In the case of the (0001-A) facet, the pDOS of the P(1) edge atoms is shown to be significantly lower (0.0573 states/eV) than the pDOS of the central (bulk) atoms (0.108 states/eV), approaching that of the P(2) bulk atoms (0.0424 states/eV). Similarly, the P(2) edge atoms acquire a much lower pDOS (0.0275 states/eV) than the bulk-like central atoms. On the other hand, in the case of the (10$\bar{1}$0) structure, the pDOS at the Fermi level of the edge P(1) and P(2) atoms, acquire similar values ~0.34 states/eV, that are close to that of bulk P(2). Hence, in the case that the (0001-A) facet dominates the termination of the nanoparticles, NMR shift distribution is expected to appear between the P(1) and P(2) NMR peaks of the bulk material, due to the intermediate pDOS values, whereas if the (10$\bar{1}$0) facet prevails, the signal intensity is expected to increase at the frequency position of the P(2) NMR peak. Importantly, all structural and electronic changes are observed to take place within a few unit cells from the surface (see Supplementary Figs. 7, 8), thus showing that relatively "thin" slabs provide reliable calculated surface electronic properties, without any interference between atoms at the fringes of the two terminating surfaces. This also explains the reason that the nanoparticles with mean size 40 nm show no profound surface NMR signal. Considering a surface shell ranging between 0.55 nm (0001-A) to 0.82 nm (10$\bar{1}$0), as explained below, and the nanoparticle size distribution shown in Supplementary Information Fig. 2d, the corresponding NMR frequency distribution between the two primary resonances at 4000 ppm and 1500 ppm is below the detection limit, as it mostly represents 1–5% of all resonating nuclei.

**Crystal facet determination of ultrasmall Ni$_2$P nanoparticles.** In the previous section it was demonstrated that downsizing of the Ni$_2$P nanoparticles significantly modifies the electronic properties of the terminating surfaces. Although the exposure of specific facets is a crucial factor in optimizing the catalytic activity of the Ni$_2$P nanoparticles, this information is not readily accessible from XRD, whilst TEM and related techniques on the one hand provide exact structural information, but they barely provide any direct information on the electron band structure. By contrast, the Knight shift reflects both the structural and electronic changes that take place at the terminating surfaces. In order to determine the

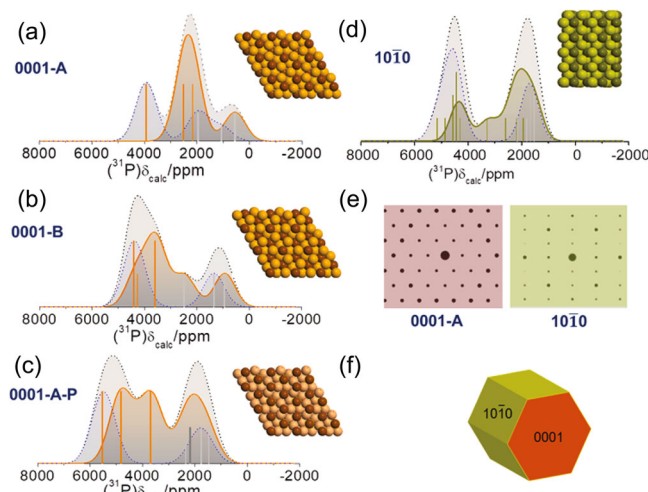

**Fig. 4 DFT-calculated** $^{31}$**P ssNMR spectra of 4 Ni$_2$P slabs with different terminating surfaces.** NMR spectra from surfaces are in orange/green color, whereas blue color spectral contribution is from the bulk-like interior. The total NMR spectra are shown with dotted lines in gray color. The superimposed orange/olive and gray color lines indicate the calculated isotropic resonances (not including lineshape effects) of the P(1) and P(2) sites, respectively. **a** $^{31}$P DFT-calculated NMR spectrum of a Ni$_2$P slab with (0001-A) termination (super-cell with 50 atoms). **b** $^{31}$P DFT NMR spectrum of Ni$_2$P with (0001-B) termination (49 atoms). **c** $^{31}$P DFT NMR spectrum of Ni$_2$P with (0001 − A − P) termination (52 atoms). The dark gray colored bar represents the isotropic resonances of the two surface P adatoms (again with no lineshape effects). **d** $^{31}$P DFT NMR spectrum of Ni$_2$P with (10$\bar{1}$0) termination (54 atoms). **e** The DFT-calculated electron diffraction patterns (EDP) of the (0001-A) and (10$\bar{1}$0) terminating slabs, which are the two most common facets in the Ni$_2$P nanoparticles. **f** Schematic presentation of Ni$_2$P nanoparticles in the P-excess limit, with the (0001) (red) and the (10$\bar{1}$0) (green) facets terminations[17].

kind of exposed facets and the way that the electronic properties vary from the surface to the core of the Ni$_2$P nanoparticles, $^{31}$P DFT- calculated NMR shifts were acquired, within the formalism of the WIEN2K code[36]. Four distinct facets were investigated: the (0001-A), which is Ni$_3$P$_2$ terminated, (0001-B) which is Ni$_3$P terminated, the (0001-A-P) with extra P adatoms on top of the (0001-A) surface, and the (10$\bar{1}$0) facet, as depicted in Fig. 4a–d, along with the DFT-calculated $^{31}$P ssNMR spectra. Orange and green shaded areas are the calculated $^{31}$P NMR spectra of the (0001) and (10$\bar{1}$0) surfaces respectively, whereas blue shaded areas

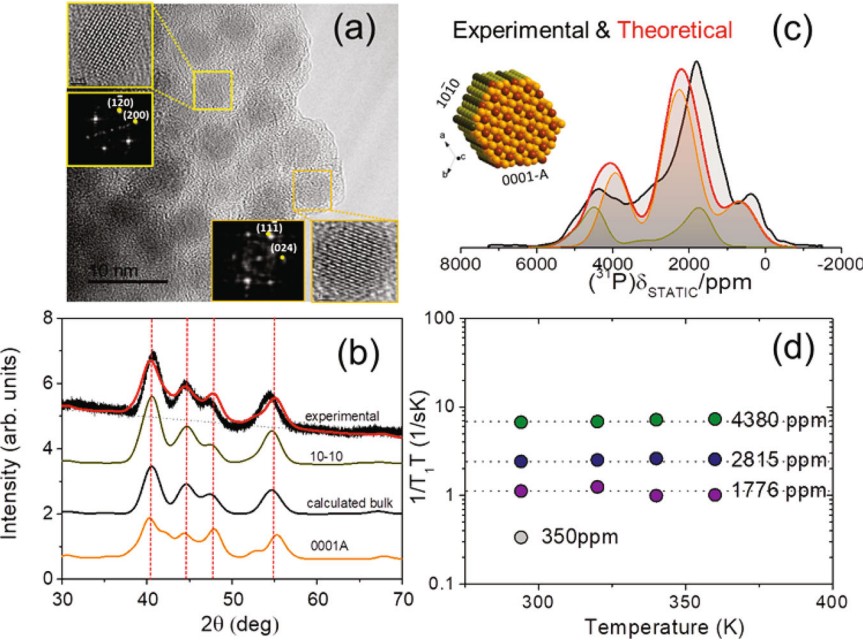

**Fig. 5 TEM, XRD and $^{31}$P ssNMR analysis of ultrafine Ni$_2$P nanoparticles with average particle size of 4.3 nm. a** TEM image of the nanoparticles exhibiting the hexagonal symmetry. Insets show magnified images of nanoparticles and the relevant EDPs, which match the calculated EDPs of the ($10\bar{1}0$) facet (Fig. 4e). Further details are provided in Supplementary Information Fig 5. **b** Experimental (black) and calculated (red) XRD patterns of the nanoparticles. The orange/green/black traces are Gaussian broadened calculated XRD simulations of the (0001-A), ($10\bar{1}0$), and bulk structures. The calculated fit (red line) is sum of the (0001-A) and ($10\bar{1}0$) patterns in ratio 1.75:1. **c** Experimental (black) and calculated (red) $^{31}$P ssNMR spectrum of the nanoparticles. The orange/green traces are the surface (0001 A)/($10\bar{1}0$) contributions in the ratio 3.67:1. The inset is sketch of the expected average facet morphology. **d** $1/T_1T$ vs. temperature for the peaks in the NMR spectrum for the ultrafine nanoparticles.

are the calculated spectra of P atoms in the central slab regions with bulk-like character. Furthermore, orange/green color and gray bars in each figure show the frequency positions of the isotropic Knight shifts of the P(1) and P(2) atoms, respectively. Similarly to the bulk Ni$_2$P, the $^{31}$P calculated Fermi-contact shielding $\sigma_{FC}$ dominates on the orbital shielding ($\sigma_{orb}$ takes values between −200 ppm to 200 ppm), while the $\sigma_{dip}$ contribution is negligible.

Figure 4a shows the calculated NMR lineshape of the (0001-A) facet. In this configuration, the surface P(1) NMR resonances are observed to shift strongly to lower shifts, acquiring values in the range 2000–3000 ppm. At the same time the surface P(2) NMR resonance shifts down to ∼700 ppm. Considering that the surface/total NMR signals integral ratio is equal to ∼0.59, the surface layer thickness is estimated to be ∼0.545 nm (total thickness of both exposed surfaces ∼1.1 nm in Ni$_2$P slab with total thickness ∼1.85 nm). Notably, the NMR lineshape matches with the relevant s-electrons pDOS distribution of Fig. 2b, showcasing the validity of Eq. (1). In case of the (0001-B) facet (Fig. 4b) the surface P(1) NMR peaks are similarly shifted to lower frequencies, however the shift is not as profound as in the case of the (0001-A) facet, apparently due to shielding of the outer P(1) sites from the P(2) sites in the terminating Ni$_3$P surface layers.

In the case of the (0001-A-P) facet (Fig. 4c) a noticeable positive frequency shift was observed for all P atoms. This can be attributed to the reduction of the Ni atoms on the layers from Ni$^{2+}$/Ni$^{3+}$ and Ni$^{+}$/Ni$^{2+}$ in order to bond with the P-adatom, leading to more unpaired electrons that are delocalizing to the s-orbitals in the vicinity of the P atoms[37]. Finally, in case of the ($10\bar{1}0$) terminating surface (Fig. 4d), the calculated NMR spectrum consists of two primary peaks flanking a low intensity resonance, in agreement with the relevant pDOS of the

s-electrons $N(E_F(|s>))$ presented in Fig. 3a, again demonstrating the prevalence of the Fermi-contact term. In this case, the surface/total NMR signal integral ratio is equal to ∼0.5, and the surface layer thickness is estimated ∼0.75 nm (total thickness of both exposed surfaces ∼1.5 nm in slab with total thickness ∼3 nm). Most importantly, the P(1) NMR resonances spread down to shifts lower than 2000 ppm, mixing with the P(2) resonances, a key observation that is experimentally verified below.

Figure 5 encapsulates the capacity of DFT-assisted $^{31}$P NMR to acquire atomic-scale information on the crystal morphology and electronic structure of the surface of ultrafine Ni$_2$P nanoparticles with a mean size of 4.3 nm. The high-resolution TEM images shown in Fig. 5a with the relevant electron diffraction patterns (EDPs) indicate the formation of hexagonally crystalized nanoparticles. Comparison of the DFT-calculated EDP of the ($10\bar{1}0$) and (0001-A) facets in Fig. 4e with the experimental EDPs in Fig. 5a unveils clearly the presence of nanoparticles exhibiting the ($10\bar{1}0$) facet (further details are provided in Supplementary Fig. 5), however, it has been extremely difficult to identify TEM images of nanoparticles exhibiting the (0001-A) facet, rendering a full facet assignment via TEM not possible. This is so, as zone-axis tilting makes facet identification in ultrasmall nanoparticles extremely difficult, as shown in Supplementary Fig. 5. Until now, the formation of terminating (0001-A) facets has been observed only by scanning tunneling microscopy maps of the Phosphorous atoms[12,15].

Similarly, XRD, does not lead to any definite conclusion. Firstly, the experimental XRD patterns of the nanoparticles are very similar with the calculated XRD of the bulk system, as clearly seen in Fig. 5b; Evidently, lattice distortions induced by the terminating surfaces in all diffraction planes in such small dimensions, broaden the experimental XRD peaks without encoding, at least in the resolution of standard XRD methods,

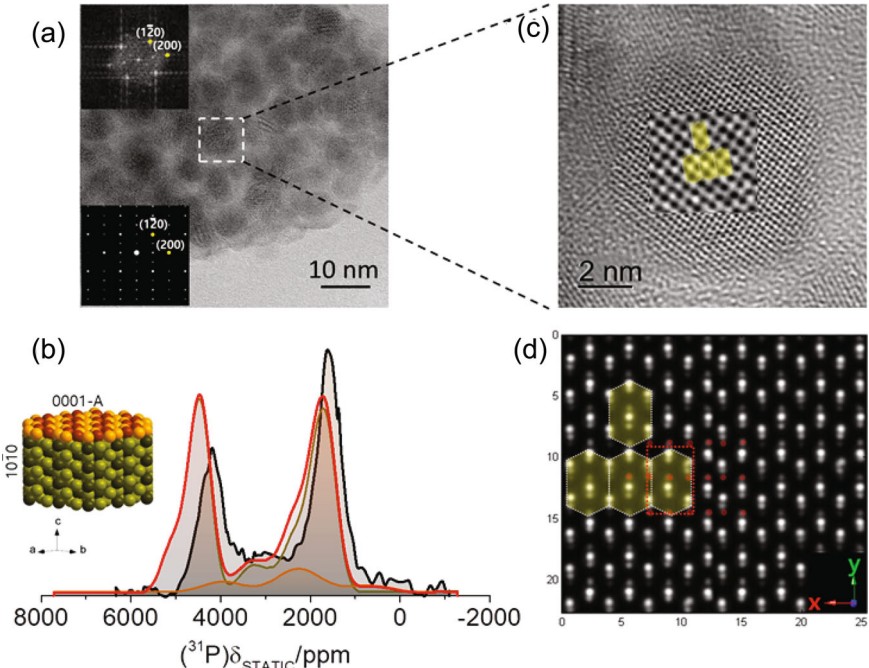

**Fig. 6 TEM, and $^{31}$P ssNMR analysis of Ni$_2$P nanoparticles of mean size 6.5 nm grown on reduced graphene oxide. a** TEM image of the nanoparticles. The upper inset shows the experimental EDP of the highlighted nanoparticle. The lower inset shows the EDP of the DFT-calculated TEM of the (10$\bar{1}$0) facet slab. **b** Experimental (black) and calculated (red) $^{31}$P ssNMR spectrum of the nanoparticles. The orange/olive color traces are the surface (0001-A)/(10$\bar{1}$0) contributions, respectively. **c** HRTEM of the highlighted Ni$_2$P nanoparticle in (**a**). The magnified region showcases the (10$\bar{1}$0) facet crystal structure along the [0001] zone axis. **d** The calculated TEM image of the DFT relaxed (10$\bar{1}$0) facet slab. Red color spots are P atoms (not observed experimentally). The red dotted rectangle is the unit cell projected onto the (**a**, **b**) plane.

any additional information for the surfaces. On the other hand, the XRD patterns calculated from the considered ultrathin slabs, which are infinitely periodic in the ab plane, imprint nicely the crystal structure of the terminating surfaces, as seen from the difference in the calculated XRD of the (0001-A) and (10$\bar{1}$0) facets. Nevertheless, efforts to simulate the experimental XRD by combining the calculated XRDs of the (0001-A) and (10$\bar{1}$0) slabs lead to significant underestimation of the 0001-A facet, I(0001 − A)/I(10$\bar{1}$0) ≤ 1.75 (Fig. 5b).

By contrast, the DFT-calculated $^{31}$P ssNMR spectrum (Fig. 5c), overlaid on the experimental spectrum, showcase unambiguously the presence of both the (0001-A) and (10$\bar{1}$0) terminating surfaces, with primarily the (0001-A) facet in ratio ∼3.67. The great advantage of NMR in comparison to diffraction techniques is that NMR is a local probe and it can discriminate between local environments within the surface and bulk irrespective of the geometry, which in case of diffraction techniques is critical at the nanoscale. The experimental $^{31}$P NMR spectrum exhibits a broad resonance at 1776 ppm and two "shoulders" at 4380 ppm and 300 ppm, accompanied by an additional feature at around 3000 ppm. It is noticed that the coherence lifetimes $T_2'$ of the P(1) and P(2) sites measured under static conditions are 244 μs and 448 μs respectively (Supplementary Fig. 16), and so some differential signal loss is expected, which might perturb the relevant signal intensities from their nominal values. However, with the appropriate experimental setup (half echo delay was set to 30 μs) the signal intensity perturbation was limited to ∼8% that is easily reckoned. The calculated NMR spectrum was optimized by fitting the intensity ratio, the Gaussian broadening, and the frequency position of the (0001-A) and (10$\bar{1}$0) facets NMR signals, in the way described in the Methods section. By taking into account the contribution of each facet into the NMR

spectrum, the surface/volume ratio according to the fit is ∼0.57. On the basis of these results and the observation of (10$\bar{1}$0) facets in the FDP analysis of the TEM images (insets in Fig. 4a), previously observed in Ni$_2$P nanorods[18], it is anticipated that ultrasmall Ni$_2$P nanoparticles exhibit the morphology shown in the inset of Fig. 5c, in agreement with theoretical predictions for nanoparticles synthesized in phosphorous excess[17].

The dominance of the surface effects in the 4.3 nm Ni$_2$P nanoparticles is furthermore evidenced by the $1/T_1T$ vs. $T$ plots (where $T$ is temperature) in Fig. 5d. First of all, these measurements show that the Korringa relation, and thus the metallic character of the nanoparticles are preserved. Besides, the enhancement of the $1/T_1T$ ratio on the two main NMR peaks of the nanosized system at frequencies 4380 ppm and 1776 ppm compared to the microcrystalline material (Supplementary Fig. 16) implies that the s-electrons pDOS $N(E_F(|s>))$ at the Fermi level is increased, following Equation S3 in Supplementary Note 5. Overall, our NMR experiments validate the DOS plots presented in Supplementary Figs. 8–10, and subsequently the shift of the Ni(1) d-electron DOS towards the Fermi level. This is important as, according to the d-band model of the catalytic activity[29], it indicates the electronic origin of the enhanced catalytic activity of the Ni$_2$P nanoparticles.

Figure 6 shows the experimental and calculated TEM and ssNMR results for Ni$_2$P nanoparticles with average size ∼6.5 nm, grown on reduced graphene oxide (r-GO). Details on the synthesis of the r-GO-supported Ni$_2$P are provided in the "Methods" section. Notably, the best fit of the theoretical NMR spectrum to the experimental spectrum occurs in case that the (10$\bar{1}$0) facet NMR signal (olive color line) overrules the (0001-A) signal, whereas a least-error simulation with the fitting procedure described above gives a ratio of the two signal intensities ∼7:1.

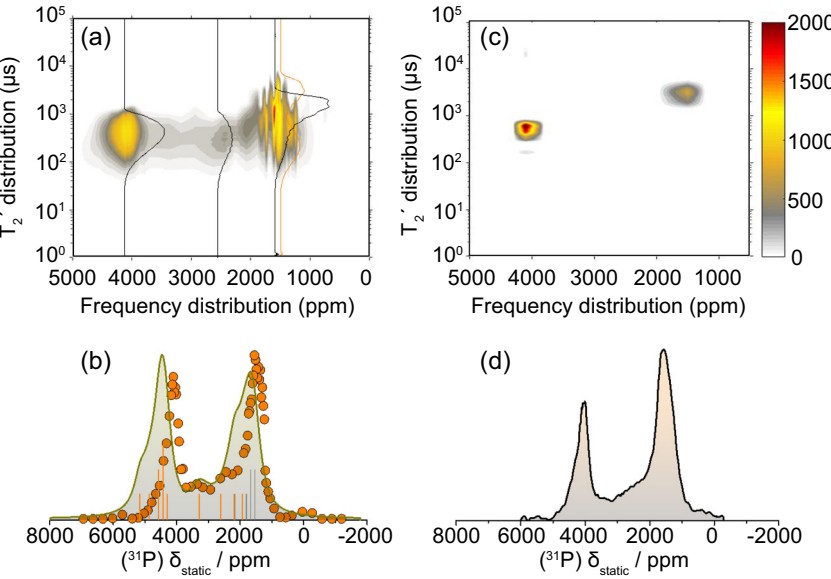

**Fig. 7 $^{31}$P NMR $T_2'$ dephasing analysis combined with DFT-calculated NMR showcasing the dominance of the ($10\bar{1}0$) facets by increasing the size of the nanoparticles. a** The $^{31}$P NMR $T_2'$ distribution as a function of the shift for the 12 nm nanocrystalline Ni$_2$P. Black cross-sections are representative $T_2'$ distributions at specific shifts which, for better visualization have rescaled intensities. **b** DFT-calculated $^{31}$P NMR spectrum (olive color line). The orange color lines are the isotropic resonances of P(1) sites, and gray lines are the isotropic resonances of P(2) sites. Orange color circles are the signal intensities calculated by inverting the CPMG spin-echo decays. **c** The $^{31}$P NMR $T_2'$ distribution as a function of the NMR frequency shift for microcrystalline (bulk) Ni$_2$P. **d** The experimental $^{31}$P ssNMR spectrum of the nanocrystalline sample.

This conclusion is supported by the high-resolution TEM (HRTEM) analysis in Fig. 6c. A great number of nanoparticles were observed to expose the ($10\bar{1}0$) surface, as shown in the high-resolution TEM in Fig. 6c; the inset with the magnified image region nearly parallel to the (0001) zone axis, displays the same crystal structure as the simulated TEM image of the DFT-relaxed ($10\bar{1}0$) facet in Fig. 6d. Furthermore, the experimental and calculated EDPs are in excellent agreement as shown in Fig. 6a. Since the ($10\bar{1}0$) termination is considered to cover mainly lateral facets[17,18], we anticipate that the nanoparticles grow preferentially along the [0001] direction. In this context, considering as mean surface layer thickness that of the ($10\bar{1}0$) facet ~0.75 nm and tiny hexagonal cylinders with size 6.5 nm, the surface layer to total nanoparticle volume ratio is estimated ~0.44, whereas the relevant $^{31}$P NMR signal intensity ratio is ~0.51, as obtained by the fit in Fig. 6b.

The preferential growth of the Ni$_2$P nanocrystals along the [0001] direction is further supported in Fig. 7, which displays $^{31}$P NMR $T_2'$ relaxation and Knight shift analysis of the 12 nm nanocrystalline Ni$_2$P sample. No information on possible facets could be revealed from the relevant XRD and TEM presented in Supplementary Figs. 1b, 4a, respectively.

Figure 7a shows the $^{31}$P NMR $T_2'$ relaxation distribution vs. frequency of the 12 nm nanocrystalline Ni$_2$P sample, acquired by inverting Carr–Purcell–Meiboom–Gill (CPMG) spin-echo trains at different resonances. In addition to the inherent spin-spin $T_2$ relaxation, additional signal dephasing arises due to coherent spin interactions, primarily the extended nuclear dipolar coupling network across the particles[38]. The measured dephasing times are referred to as coherent lifetimes $T_2'$. Details on the inversion of the CPMG decay curves are presented in the Supplementary Information Note 4. For reasons of comparison, the relevant $T_2'$ distribution vs. frequency plot of the bulk Ni$_2$P is presented in Fig. 7c. Remarkably, in the nanocrystalline material, the $T_2'$ distribution of the P(1) sites, initially at ~4200 ppm, is observed to "spread" down to ~1700 ppm, i.e., the frequency of the P(2) site. The $T_2'$ distribution analysis at characteristic frequency

positions (black lines), as well as the $T_2'$ analysis of the relevant CPMGs presented in Supplementary Information Fig. 15, show clearly the presence of two $T_2'$ components at frequency shift ~1700 ppm, which may be assigned to overlapping P(1) and P(2) NMR signals. This finding ties excellently with the spread of the calculated isotropic resonances of surface P(1) sites, as seen in Fig. 7b. Furthermore, the calculated ($10\bar{1}0$) NMR spectrum with Lorentzian broadening ~70 kHz and the experimental spectrum are in good agreement; however, one should notice that in addition to the bulk resonances inherent in the ($10\bar{1}0$) calculated spectrum, a contribution of up to 20% more of the bulk resonances is expected, if we consider that the nanoparticles are hexagonal nano-cylinders with mean size ~12 nm. The experimental spectrum in Fig. 7b was acquired by plotting the calculated signal intensity (i.e., the nuclear magnetization M$_0$) from the CPMG inversions as a function of frequency. No reasonable fit was possible by considering eventual contribution from the (0001-B), or the (0001-A-P) facets, whereas even the (0001-A) facet contribution is not necessary to simulate the experimental spectrum. These results are evidence that by increasing size lateral ($10\bar{1}0$) facets dominate, and nanoparticles acquire the shape illustrated in Fig. 6b.

## Discussion

$^{31}$P DFT-assisted solid-state NMR combined with HRTEM and XRD crystal structure analysis were successfully employed to identify the individual crystal facets of the terminating surfaces of Ni$_2$P nanoparticles. It is shown that mainly two facets contribute to the experimental results, the (0001-A) and the ($10\bar{1}0$) facets, with the former being the dominant terminating surface of the ultrathin nanoparticles. Notably, as the nanoparticles grow in size, the NMR signal from the ($10\bar{1}0$) facets appears to prevail over the one from the (0001-A) facets, indicating that nanoparticles grow along the [0001] zone axis, exposing predominantly lateral ($10\bar{1}0$) surfaces. Our calculated NMR spectra are part of extensive DFT calculations, which show that the exposed (0001-A) and ($10\bar{1}0$) facets shift the Ni electron d-bands towards the

Fermi level, indicating augmented catalytic activity according to the d-band model of catalysis[29]. This conclusion is important for many applications; For example, in a recent study of nanostructured phosphides nanosheets it was shown that the rise of $E_F$ is responsible for the significant promotion of oxygen evolution catalysis[39]. It is noticed that since NMR is a local probe, the method is applicable even for particle sizes of a few nanometers, where most other techniques for structural characterization fail. This makes the method an important tool in understanding the complex structural and electronic properties of important nanocatalysts.

## Methods

**Synthesis of Ni₂P nanoparticles**. The microcrystalline (bulk) Ni₂P sample was procured from SIGMA-ALDRICH. Ni₂P and Ni₁₂P₅ nanoparticles were synthesized using a slight modification of previously reported liquid phase approach based on solvothermal phosphidation using tri-octylphosphine (TOP), as phosphorous source in oleyl amine enviroment[40]. The Ni₂P nanoparticles were synthesized in primary amine (oleyl amine)/tertiary amine (trioctyl amine) 1/9 volume ratio under high phosphorous excess, with P/Ni²⁺ molar ratio = 2.8 at 300 °C. Aiming to enhance the phosphidation process and synthesize the hexagonal Ni₂P phase in a controllable manner, we have replaced part of the primary amine (oleyl amine), which binds strongly to the metallic surfaces, with a tertiary amine (trioctyl amine), which due to the high steric effect originating from the presence of three alkyl groups binds weakly to the nanoparticle surface, but work in a similar way as oleyl amine as a coordination solvent through the electron-pair donor properties. All the reaction steps took place under inert atmosphere. The reaction mixture changed progressively from green to dark green and finally to black indicating the formation of the phosphide nanoparticles. The nanoparticles were precipitated, after cooling at room temperature, by adding ethanol and separated by centrifugation. This precipitation/separation procedure was repeated twice in order to remove the excess of the organic molecules and reaction by-products.

The Ni₂P/r-GO hybrids were synthesized following similar methodology, with sonochemical GO exfoliation before the Ni₂P formation, as described in ref. [41]. The main difference in this case is the replacement of octadecene with tri-octylamine. The tri-octylamine, in contrast with octadecene, functionalize more efficiently, and consequently exfoliate better, the GO flakes, and additionally the competitive adsorption with the primary amine, due to steric effects, on the nickel surface enhance the phosphidation process leading to the formation of the phosphorous-rich Ni₂P phase. In particular, GO were first dispersed in the aliphatic amines (oleyl amine/tri-octylamine) mixture and probe sonicated in order to exfoliate them, following by the drop-wise addition of the Ni(acac)₂ solution in a similar aliphatic amines mixture which additional includes the phosphorous source, (TOP). The reaction mixture was then removed from the sonication source and heated up to 300 °C for 2 h under a nitrogen blanket. The products separated first by centrifugation and washed twice with a 1/1 ethanol-hexane mixture in order to remove the excess of the organic solvents and reaction by-products, and finally stored in CHCl₃.

In all synthesized materials after washing no amine/TOP traces chemisorbed on the nanoparticle grains were observed, as exemplified in the FT-IR spectrum of the 12 nm sample in the Supplementary Information Fig. 4b.

**Solid-state NMR**. The ³¹P MAS experiments were performed with a 4 mm HXY triple-resonance probe, at 14 kHz MAS on a Bruker 400 Avance-III spectrometer operating at a ³¹P Larmor frequency of 161.976 MHz. Spectral acquisition was done with a double adiabatic spin-echo sequence (DAE) with a 5.0 μs π/2 excitation pulse length, corresponding to an RF field of 50 kHz, followed by a pair of rotor-synchronized tanh/tan short high-power adiabatic pulses (SHAPs)[42,43] of 71.43 μs length and 5 MHz transmitter frequency sweep. Three subspectra acquired using the variable offset cumulative spectroscopy (VOCS) method were summed[44]. Chemical shifts were referenced to H₃PO₄ 85% wt at 0 ppm. The use of adiabatic pulses (DAE pulse sequence), which have the capability to generate uniform rotations of the nuclear magnetization even when applied radiofrequency field is highly inhomogeneous[45], compensates changes in the rf field amplitude, when penetrating the microcrystalline grains of bulk Ni₂P. The penetration depth at the resonance frequency of 162.066 MHz is ~42 μm, whereas microcrystalline Ni₂P grains have size distribution between 2 and 120 μm.

The ³¹P static NMR experiments were performed at magnetic field of 9.4 T using a home-built broadband coherent pulsed NMR spectrometer. An Oxford furnace was employed for high temperature measurements. The NMR spectra were obtained using a home-built probe and employing the standard Hahn spin-echo pulse sequence, while varying the frequency across the NMR line. The π/2 pulse length of the rf pulses was set to 3.5 μsec, whereas NMR lineshape measurements were performed with a Hahn-echo π/2-τ-π pulse sequence with τ = 30 μs.

**XRD**. The XRD patterns were acquired with a Siemens D500 powder diffractometer with Cu-Kα radiation (λ = 1.5418 Å). In order to enhance the intensity

and resolution of the diffraction maxima, a Bragg–Brentano focusing geometry was used. X-ray patterns were recorded at an angular range (2θ) of 20–100° and at a scanning rate 0.5°/min. The Rietveld refinement of obtained powder XRD pattern was carried out using the FULLPROF program software. Refined parameters include: overall scale factor, background (BGP), lattice parameters, atomic positions and orientation. DFT-Calculated XRD patterns were acquired using the VESTA software[46].

**TEM**. TEM studies were carried out on a JEOL JEM-2100F electron microscope and a FEI Tecnai G20, both operating at 200 kV. For this analysis, approximately 2 mg of each sample was dispersed in 20 ml of high purity cyclohexane (Merck, 99.9%) via sonication (~15 s); then a drop of each suspension was deposited on carbon-coated grids (400 mesh) covered with thin amorphous carbon film (lacey carbon). To avoid contamination, specimens were inserted in the TEM immediately following preparation. Bright field images were collected at several magnifications in order to observe structure and size homogeneity. Calculated HRTEM images were obtained with the help of the MacTempasX software, Total Resolution LLC. The simulated electron diffraction pattern images were obtained with the SingleCrystal software (CrystalMaker Software Ltd.).

**DFT calculations**. DFT calculations were performed using the QUANTUM ESPRESSO Simulation Package code[47]. The Perdew-Burke–Ernzerhof exchange-correlation functional with the generalized gradient approximation[48] was used. The Projector-Augmented Wave approach (PAW)[49] was used together with plane-wave basis sets. Since Ni₂P in both bulk and nanocrystalline forms is metallic there is no need for using hybrid functionals such as HSE06[50], or GGA + U[51] to describe Ni d-bands. In case of the bulk system, calculations were performed by inserting in the DFT calculations the lattice parameters from the Rietveld analysis. The kinetic energy cutoff of 480 eV was used and the Brillouin zone was sampled by a 12 × 12 × 12 Monkhorst-Pack mesh[52]. The z-axis was taken as normal to the surface.

The Ni₂P nanoparticles were modeled with a Ni₂P super-cell comprised of 5–11 Ni₂P unit cells (successive Ni₃P₂ and Ni₃P layers), terminated with Ni₃P₂ in case of the (0001-A) slab, and Ni₃P in case of the (0001-B) slab. The (0001-A-P) structure was taken from ref. [17], whereas the (10̄10) slab comprised of a 54-atoms super-cell was designed with the VESTA software. All structures were relaxed using the Broyden–Fletcher–Goldfarb–Shanno algorithm[53]. The kinetic energy cutoff was set 480 eV and a 12 × 12 × 1 Monkhorst-Pack mesh was used.

NMR Knight shift calculations were performed on bulk Ni₂P, and the 4 different Ni₂P slabs by using the full-potential linearized augmented plane-wave method, as implemented in the Wien2k DFT software package[36]. The k-mesh convergence was checked up to 100,000 points for the bulk materials and up to 5000 points for the slabs. Other computational parameters, such atomic sphere radii as well as potentials and wave functions inside the atomic spheres, were as set by Wien2k defaults. The plane-wave basis set size was determined by setting $RK_{max} = 8$, whereas for presented results we have used the LDA approximation[54]. All structures, including the bulk material were structurally relaxed. In case of the bulk material the initial lattice parameters were taken from the Rietveld analysis of the XRD pattern. The calculated spectra fits in Figs. 5c, 6b, 7b were obtained by letting the frequency position, Gaussian broadening, and intensity ratio of each facet NMR signal components to vary during fitting. Small percentage fitting components were excluded and then refitted. At the end of the procedure, the initial frequency position of each calculated component was restored. The presented calculated spectra exhibit the best fit of all NMR facet combinations.

## Data availability

The authors declare that the data supporting the findings of this study are available within the article and its Supplementary Information file. Extra data are available from the corresponding authors upon reasonable request.

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

## Acknowledgements

W.P., J.P.C., and A.J.P. were supported by the Swedish Research Council (project no. 2016-03441) and the Swedish National Infrastructure for Computing (SNIC) through the center for parallel computing (PDC), project number 2019-3-500. N.P., V.T., M.K., M.F., and G.P. acknowledge support by the project MIS 5002567, implemented under the "Action for the Strategic Development on the Research and Technological Sector", funded by the NSRF 2014-2020 and co-financed by the European Union and Greece. Part of the DFT work was performed using computational resources of the Research Computing Department at Khalifa University. X.L., K.P., G.P., and Y.A. would like to acknowledge the support of Khalifa University of Science and Technology Award No. RC2-2018-024.

## Author contributions

A.J.P., W.P., Y.A., and G.P. conceived and designed the experiments, performed theoretical analysis, and did the majority of paper writing (with additional contributions from all coauthors). MAS NMR experiments and data analysis were performed by W.P., J.P.C., A.J.P., and V.K. Frequency sweep static NMR and relaxation studies were performed by W.P., N.P., M.K., and M.F. DFT calculations and comparison with experiments were performed by W.P., Y.A., J.P.C., and G.P. Ni$_2$P nanoparticle preparation and characterization was performed by X.L., K.P., and V.T. HRTEM experiments and simulations were performed by J.B.L., S.L., C.Y.K., H.J.K., and M.K.

## Funding

## Competing interests

The authors declare no competing interests.
