## [Peer Review File · Nature Communications]

REVIEWER COMMENTS

Reviewer #1 (Remarks to the Author):

This manuscript investigates the facets of Ni₂P nanoparticles using the ³¹P solid-state NMR method, combining the HRTEM, XRD, and DFT simulations. The ssNMR would be a meaningful tool to identify the individual crystal facets of the terminating surfaces, for the quantum dot crystals, though it has been reported to determine the distinct surface facets. The characteristic ssNMR spectra were presented as scaling down the Ni₂P particle size to the quantum-dot level, and the DFT results could prove the correlation of the spectra-crystal facet terminating surfaces.

I think this article interesting, but more solid experimental evidence seems to me is needed to support the authors' ideas. I believe there are a few points that require further discussion.

- The 0001-A dominant terminating surface was analyzed by ssNMR, TEM, and XRD in the ultrathin Ni₂P nanoparticles with an average particle size of 4.3 nm. The {0001} facet is not be observed in TEM images. Further, it seems that the experimental and calculated XRD patterns fail to match very well. Please confirm the fitting results and reliability.

- No chemical shift presents in ssNMR spectra of Fig 1a and Fig 2a (commercial and 40 nm Ni₂P). The ratio of {0001} and {10-10} facets changes along with different size in Ni₂P particles? Is the ssNMR sensitive toward the variation of the dominated facet in Ni₂P crystal?

- It has been known that the GGA functional always can not describe the localization of 3d electron of the transition metals (e.g. Fe, Cr, Co, Ni) correctly, authors should use the HSE06 or at least GGA+U approach for Ni₂P in DFT calculations.

- The ideal ultra-small hexagonal crystal cell had been presumed to simulate the ³¹P ssNMR, and only {0001} and {10-10} facets were considered. In fact, other facets could be also exposed and many defects occur on the particle surface, scaling down to quantum-size. Their effects on NMR spectra might not be ignored and were not discussed in the manuscript.

- The Ni electron d-bands should be calculated on the serial Ni₂P nanoparticles and their changes should be analyzed, as their importance towards the catalytic activity.

- Crystallographic descriptors should be written according to the routines, such as crystalline facets, directions, and zone axis, etc.

- In the fourth paragraph, the authors emphasize that "Most importantly, the nanoparticles appear to have a thin rod-like shape with lateral 1010 surfaces that become more prevalent by increasing size of the nanoparticles. At the same time, on both terminating surfaces, the Ni d-electron bands shift towards the Fermi level, verifying that the highly enhanced catalytic activity of the Ni₂P nanoparticles to a great extent originates from the changes in the surface electronic properties." Any rod-like morphology of Ni₂P was demonstrated in the manuscript? The catalytic activity of Ni₂P was not measured in this work.

- Some references may be missed, like "The black (Ni₂P) and green (Ni₁₂P₅) XRD patterns are simulations acquired from literature cif files that..." in Supplementary Figure 1.

Reviewer #2 (Remarks to the Author):

The manuscript by Papawassiliou et al. describes the analysis of the facets of Nanosized Ni₂P based on DFT and NMR measurements as well as TEM and XRD.

This is a potentially interesting contribution due to the important role of facets in many applications and mostly catalytic activity.

Nevertheless, there are several issues that must be addressed, which are essential for supporting the presented interpretation of the NMR data:

(i) The resolution of the NMR spectra, especially in the case of the nanoparticles, is relatively low and the different resonances spread over a wide range of shifts.

The assignment of the NMR spectra is mostly based on DFT calculations of the different surface models. However, it seems that most of the calculated models shown in figure 4 could be used to fit the experimental data, as they all span roughly the same frequency range.

Moreover, they can be combined to fit the spectra with varying contributions. This was done for example in Figure 5c. This specific choice works well (at least visually) for the 4.3nm particles but there are very large deviations in the case of the 12nm particles. Why would there be such a difference in the fits of the two particles if the calculated shifts fit the models?

Why can't the other models be used in the same way and what in the experimental data suggest that they are not suitable?

(ii) NMR is usually quantitative and this property must be used to support the suggested models for the particles shape: for example what should be the relative intensity contributing to the different facets vs. bulk? this can be calculated from the models and compared to the experimental spectra considering the assignment of the resonances.

Are the spectra quantitative? how about RF skin depth effects in metallic particles? perhaps for the nm sized particle this is negligible but for micron size should have an effect that would change the surface/bulk ratio.

Data was acquired with echo sequences: what was the echo delay? what are the T₂ values of the different sites? variation in transverse relaxation may change the relative intensity, make the relative intensities in the spectra meaningless which will make comparison of bulk vs. surface challenging.

(iii) Page 4 - description of the calculated NMR spectra vs. experimental 'an excellent match with the experimental ³¹P NMR data' - there is a good match between the isotropic shifts in experiment and calculation but the anisotropies as reflected in the spinning sidebands are not reproduced well in the calculations. This should be addressed in the text and the values of the isotropic resonances and shielding parameters (exp and calc) should be provided in the SI.

(iv) Calculated DFT results plotted on top of experimental data in Figure 5-7 with certain linewidth and weight for the distributions. Histograms of the isotropic resonances should be provided for all models. In the description of the fits, the width added to the calculated resonance should be detailed as well as the relative intensity.

(v) Are the nanoparticles capped by ligands? if so can the ligand affect the measured shift?
Will ^1H - ^{31}P CP experiments be helpful in assigning the surface resonances? or the metallic nature of the particle prevents these?

minor comments:

*figure 13 SI - add legend to the figure. The sequence in the caption does not seem right for T1 measurements, seems like an echo sequence (90-T-180)

*page 3, second paragraph, remove 'induces'

*page 6, second paragraph, line 6, remove 'are'

*page 10, 6th row from bottom: remove 'is the'

We are grateful to the reviewers for the received feedback, which we feel have led to substantial improvement of our manuscript. We have now revised the manuscript according to the points raised in their reports.

Major revisions in the article are:

- Calculated isotropic frequencies and relevant intensities for the P(1) and P(2) sites (separately for each kind) have been included in the revised Figure 4.
- Improved fits and justification of the contribution of calculated 0001-A and $10\bar{1}0$ NMR signal components are included in the revised Figures 5 and 6.
- New experimental results have been included. Specifically, the T_2' distribution vs. frequency for the microcrystalline (bulk) and the nanocrystalline 12 nm Ni_2P samples are presented in Figures 7a,c. The best fit of the calculated NMR spectrum on the experimental M_0 vs. frequency spectrum for the 12 nm sample is presented in Figure 7b.
- New Supplementary Figures 4, 15, and 16 have been added.

Below, please find the original reviewer comments in blue colour and the authors' point-by-point responses in black. All changes outlined in the text below are highlighted in yellow colour in a highlighted version of the revised manuscript. References supporting our arguments are added at the end of text.

Reviewer #1 (Remarks to the Author):

This manuscript investigates the facets of Ni_2P nanoparticles using the ^{31}P solid-state NMR method, combining the HRTEM, XRD, and DFT simulations. The ssNMR would be a meaningful tool to identify the individual crystal facets of the terminating surfaces, for the quantum dot crystals, though it has been reported to determine the distinct surface facets. The characteristic ssNMR spectra were presented as scaling down the Ni_2P particle size to the quantum-dot level, and the DFT results could prove the correlation of the spectra-crystal facet terminating surfaces. I think this article interesting, but more solid experimental evidence seems to me is needed to support the authors' ideas. I believe there are a few points that require further discussion.

We would like to thank the reviewer for their positive opinion regarding the novelty of the experimental work presented and his valuable comments, which have given us the opportunity to significantly improve the discussion on the main issues raised.

The 0001-A dominant terminating surface was analyzed by ssNMR, TEM, and XRD in the ultrathin Ni_2P nanoparticles with an average particle size of 4.3 nm. The $\{0001\}$ facet is not be observed in TEM images. Further, it seems that the experimental and calculated XRD patterns fail to match very well. Please confirm the fitting results and reliability.

Ni_2P slabs with the 0001A and $10\bar{1}0$ terminations were generated from a cif file provided by ref. 1, and subsequently DFT geometry optimized. The relaxed structures were then used to obtain calculated XRD patterns, TEM images, and NMR spectra of the relevant facets.

In case of TEM the $10\bar{1}0$ facet could be nicely identified, however the reviewer is correct that it was not possible to identify the 0001-A facet. One reason for this is that zone-axis tilting makes facet identification in ultrasmall nanoparticles extremely difficult, shown in the revised Supplementary Figure 5 in page 6.

XRD methods are in principle not in a position to determine the facet topology of ultrasmall nanoparticles. This was explicitly stated in the article, but we acknowledge that the relevant paragraph was not clearly written.

Firstly, the experimental XRD patterns of the nanoparticles are very similar with the calculated XRD of the bulk system (3D periodic), as clearly seen in Figure 5b; evidently, lattice distortions induced in such small nanoparticles by the terminating surfaces in all diffraction planes just broaden the experimental XRD peaks, without encoding any information about the terminating surfaces. On the other hand, the calculated XRD patterns produced by the ultrathin slabs, which are infinitely periodic in the ab plane, imprint excellently the crystal structure of terminated surfaces, as seen from the difference in the calculated XRD of the 0001-A and $10\bar{1}0$ facets in Figure 5b. However, efforts to simulate the experimental XRD by combining the calculated XRDs of the 0001-A and $10\bar{1}0$ slabs lead to significant underestimation of the 0001-A facet. The inadequacy of the simulated XRDs to describe experimental XRDs of the nanoparticles is due to the fact that in case of diffraction techniques, differences in geometry become very critical at the nanoscale.

This handicap is not present in the case of local techniques such as TEM and NMR, both methods probe directly the surface, whereas NMR may distinctly resolve in addition to the surface the bulk-like interior. Therefore, calculations on nanosized slabs are an excellent theoretical tool to acquaint the nanoparticles surface topology with TEM and NMR methods.

The relevant text is in page 13 in the last paragraph and continues in page 14 of the revised manuscript.

No chemical shift presents in ssNMR spectra of Fig 1a and Fig 2a (commercial and 40 nm Ni_2P). The ratio of {0001} and {10-10} facets changes along with different size in Ni_2P particles? Is the ssNMR sensitive toward the variation of the dominated facet in Ni_2P crystal?

1) The NMR signal from the surface of the 40 nm sample is invisible because the number of resonating nuclei in the surface shell is very small in comparison to the total number of resonating nuclei (low surface area). Specifically, considering (i) the nanoparticle size distribution of this sample (Supplementary Figure 2d), and (ii) the surface shell thickness, ranging from 0.55 nm (0001A) to 1 nm ($10\bar{1}0$), as calculated and explained in the revised version of the article, the percentage of resonating nuclei in the surface of the nanoparticles, with frequency between the two resonance peaks (i.e. 4000 ppm and 1500 ppm), is extremely small (~ 0.01 to 0.04%). The small bulges at the two peak resonances might be related with such surface states, but this is too vague to consider. By decreasing particle size the percentage of the resonating nuclei in the surface shell increases rapidly (to overall 0.4% of the total number of resonating nuclei for the 12 nm sample). The relevant text has been added in page 11.

2) The revised Figures 5c found in page 15, revised Figure 6b found in page 17, and revised Figure 7c found in page 18. Specifically, in the case of the 4.3 nm nanoparticles NMR spectrum, the 0001-A NMR signal component is more pronounced in comparison to the $10\bar{1}0$ facet by a ratio of ≈ 3.67 . This is in stark contrast to Figure 6b, where the NMR lineshape is dominated by the $10\bar{1}0$ facet growing laterally to the 0001-A facet, in a ratio of ≈ 7 . This is discussed in more detail in page 14 and page 16 of the revised main manuscript.

It has been known that the GGA functional always cannot describe the localization of 3d electron of the transition metals (e.g. Fe, Cr, Co, Ni) correctly, authors should use the HSE06 or at least GGA+U approach for Ni_2P in DFT calculations.

The referee is in principle right, the application of hybrid functionals such as HSE06, or the use of GGA+U is superior to GGA in transition metal compounds, provided that materials are either insulating/semiconducting^{2,3}, or their d electrons are shared between metallic and localized states, with strongly correlated properties (e.g. hole doped Mott insulators⁴). For example, simple GGA calculations erroneously predict insulating Fe₃O₄ and FeO to be metals, whereas by implementing HSE03 and GGA+U functionals they are found correctly to be insulators^{5,6}.

However, in typical metallic systems, such as Ni₂P, the use of a hybrid functional like HSE06 or PBE0² is expected to overestimate severely the bandwidth and are thus considered to be inappropriate for metallic systems^{7,8}. Similarly, GGA+U is excellent for transition metal insulators, semiconductors and strong electron correlated systems, however they are not appropriate for simple metals, where the Hubbard correction term significantly distorts the d-band around the Fermi level.

The appropriateness of GGA to calculate energies of metallic systems is nicely demonstrated in ref. [9], which compares experimental results with DFT calculations in 80 different systems. As far as we know, until now all published DFT calculations in Ni₂P and other metallic phosphides use GGA/LDA functionals (selected examples are given in references¹⁰⁻¹⁶).

An explanation to this has been added in page 22 in the DFT calculations section of the Materials and Methods section.

The ideal ultra-small hexagonal crystal cell had been presumed to simulate the ³¹P ssNMR, and only {0001} and {10-10} facets were considered. In fact, other facets could be also exposed and many defects occur on the particle surface, scaling down to quantum-size. Their effects on NMR spectra might not be ignored and were not discussed in the manuscript.

We would like to thank the reviewer for this comment because the discussion for considering the 0001-A and 10 $\bar{1}$ 0 facets was not complete in the previous version of the article. This gave us the opportunity to enrich the relevant text. There are three pillars of argument:

(1) The Synthetic procedure used:

By reducing size a variety of facets have the possibility to grow, however depending critically on temperature, the chemical environment, duration of the various reaction steps, etc. At the temperature used (300°C, please see Materials and Methods section), ab-initio atomistic thermodynamic studies¹⁴ have shown that the most stable surfaces are the 0001 the 10 $\bar{1}$ 0 and the 11 $\bar{2}$ 0. Most important, the synthesis was performed in P excess (i.e. in the P-rich regime), promoting further the formation of the 0001-A and the 10-10 facets¹⁴. Details on the exact synthesis procedure are provided in the revised Materials and Methods section in page 20, paragraph 1. Detailed discussion is regarding facet growth under P-excess is provided in the first paragraph of page 3, and the last paragraph of page 6 of the revised manuscript.

(2) The fitting procedure does not support the presence of the 0001-B and 0001-A-P NMR lineshape components. The calculated NMR spectra in the revised Figures 5c, 6b, and 7b, are the best fit spectra by probing various combinations of facets. Fits were obtained by letting the frequency position, Gaussian broadening, and intensity ratio of each facet NMR signal components vary during fitting. Small percentage fitting components were excluded and then refitted. At the end of the procedure, the initial frequency position of each calculated component was restored. The relevant discussion has been added in page 19 of the revised main manuscript. Details of the fitting process are presented in page 23 of the revised main manuscript.

(3) The T_2' distribution vs. resonance frequency in the revised main manuscript Figure 7a, provides unambiguous evidence that the surface P(1) NMR resonances shift down to 1700 ppm. This is in agreement with the DFT calculated shift of the surface P(1) resonances of the $10\bar{1}0$ facet, as shown in Figures 4d, and 7b of the revised main manuscript.

The Ni electron d-bands should be calculated on the serial Ni_2P nanoparticles and their changes should be analyzed, as their importance towards the catalytic activity.

The Ni electron d-bands are presented for all cases in the SI. Supplementary Figures 5a and 5b present the electron d-bands of bulk Ni_2P . The original supplementary Figures 6a and 6b in page present the electron d-bands of bulk Ni_2P . The original supplementary Figures 8a, 8b and 9a, 9b present the electron d-bands for the slabs representative of nanoparticles of 4.3nm and 12nm, respectively. The possibility of catalytic enhancement, according to the d-band theory - is directly tied to the shift of the d-band towards the Fermi level¹⁷. As an example, in the case of oxygen evolution reaction, lower energy of the d-band center means less likelihood of oxygen-containing intermediates to be bound to the Ni_2P surface, as the antibonding states are filled with more electrons when the adsorption process occurs^{17,18}. It is thus key to move the d-band center to higher energies, which is achieved in the case of Ni_2P nanoparticles as implied by the exposed 0001-A and the $10\bar{1}0$ facets. This has been emphasized in page 15 of the revised main manuscript as well as in the discussions section in page 20 of the revised main manuscript.

Crystallographic descriptors should be written according to the routines, such as crystalline facets, directions, and zone axis, etc.

This has been addressed throughout the text of the revised main manuscript. In addition, crystallographic descriptors have now been added to the relevant Figures both in the revised main article and the SI (see Figures 5a, 6a, and SI Figure 5.)

In the fourth paragraph, the authors emphasize that “Most importantly, the nanoparticles appear to have a thin rod-like shape with lateral $10\bar{1}0$ surfaces that become more prevalent by increasing size of the nanoparticles. At the same time, on both terminating surfaces, the Ni d-electron bands shift towards the Fermi level, verifying that the highly enhanced catalytic activity of the Ni_2P nanoparticles to a great extent originates from the changes in the surface electronic properties.” Any rod-like morphology of Ni_2P was demonstrated in the manuscript? The catalytic activity of Ni_2P was not measured in this work.

As stated above, synthesis was done in the P-rich regime; this predicts theoretically (ref. 14) cylindrical shape with hexagonal symmetry (tiny hexagonal cylinders), with frontal facets the 0001 and lateral $10\bar{1}0$. The presence of the two facets, i.e. 0001-A and $10\bar{1}0$ has been verified with NMR (the $10\bar{1}0$ also with TEM). Since the relative intensity of the signal component originating from the terminating $10\bar{1}0$ facet increases with size, this is clear evidence that the nanoparticles grow along the 0001-A direction, exposing more $10\bar{1}0$ surface area. The relevant clarification is written in page 3, in the first paragraph of the revised main manuscript.

The reviewer is right to say that we have not measured the catalytic properties. Our DFT calculations support the experimentally observed catalytic enhancement in several publications. To avoid any misunderstanding, we have rephrased the relevant text in the article to be found in the discussions section (see above).

Some references may be missed, like “The black (Ni_2P) and green (Ni_{12}P_5) XRD patterns are simulations acquired from literature cif files that...” in Supplementary Figure 1.

The sources of the CIF files have now been added, in page 20 of the revised SI.

Reviewer #2 (Remarks to the Author):

The manuscript by Papawassiliou et al. describes the analysis of the facets of Nanosized Ni₂P based on DFT and NMR measurements as well as TEM and XRD. This is a potentially interesting contribution due to the important role of facets in many applications and mostly catalytic activity. Nevertheless, there are several issues that must be addressed, which are essential for supporting the presented interpretation of the NMR data: We would like to thank the reviewer for their comments. They led to the addition of important information, such as the T_2' vs. frequency plots in Figure 7a,c, as well as overall significant improvement of the article .

Comments/Questions:

(i) The resolution of the NMR spectra, especially in the case of the nanoparticles, is relatively low and the different resonances spread over a wide range of shifts. The assignment of the NMR spectra is mostly based on DFT calculations of the different surface models. However, it seems that most of the calculated models shown in figure 4 could be used to fit the experimental data, as they all span roughly the same frequency range. Moreover, they can be combined to fit the spectra with varying contributions. This was done for example in Figure 5c. This specific choice works well (at least visually) for the 4.3nm particles but there are very large deviations in the case of the 12nm particles. Why would there be such a difference in the fits of the two particles if the calculated shifts fit the models? Why can't the other models be used in the same way and what in the experimental data suggest that they are not suitable?

Indeed, NMR spectra of nanoparticles with sizes of a few nm usually present with low resolution, due to structural changes of the particles, strains, and defects on their extended surfaces. Nevertheless, the contribution of the various facets on the NMR spectra can be evaluated by appropriate consideration of the DFT-calculated NMR spectra. The reasons that we have considered the 0001-A and 10 $\bar{1}$ 0 facets are as follows:

(1) Synthesis conditions: The Ni₂P nanoparticles were synthesized in primary amine (oleyl amine) /tertiary amine (trioctyl amine) 1/9 volume ratio under high phosphorous excess, with P/Ni²⁺ molar ratio = 2.8 at 300 °C (please see revised Materials and Methods). Ab initio atomistic thermodynamic studies have shown that the most stable surfaces grown at this temperature and in P-excess are the 0001 and the 10 $\bar{1}$ 0 facets¹⁴. The relevant discussion about the synthesis conditions is provided in the revised Materials and Methods section of the main manuscript in page 20. Additional discussion regarding the facet growth under P-excess is found in page 3 in the first paragraph and in the last paragraph of page 6 of the revised main manuscript.

(2) The calculated spectra fits in the revised Figures 5c, 6b and 7b were obtained by letting the frequency position, Gaussian broadening, and intensity ratio of each facet NMR signal components to vary during fitting. Small percentage fitting components were excluded and then refitted. At the end of the procedure, the initial frequency position of each calculated component was restored. The presented calculated spectra exhibit the best fit of all NMR facet combinations. Fitting details have been added in the materials and methods section in page 3 of the revised main manuscript. The relevant discussion has been added in page 14 of the revised main manuscript.

(3) The T_2' distribution vs. frequency in the revised Figure 7a show a substantial shift of the surface P(1) resonances down to 1700 ppm. This matches primarily with the $10\bar{1}0$ facet as shown in Figure 7b, excluding the presence of the 0001-B, and 0001-A-P facets, where surface P(1) resonances shift very little. **The relevant Figure 7 has been added in page 19 of the revised main manuscript.**

(ii) NMR is usually quantitative and this property must be used to support the suggested models for the particles shape: for example what should be the relative intensity contributing to the different facets vs. bulk? this can be calculated from the models and compared to the experimental spectra considering the assignment of the resonances. Are the spectra quantitative? how about RF skin depth effects in metallic particles? perhaps for the nm sized particle this is negligible but for micron size should have an effect that would change the surface/bulk ratio. Data was acquired with echo sequences: what was the echo delay? what are the T2 values of the different sites? variation in transverse relaxation may change the relative intensity, make the relative intensities in the spectra meaningless which will make comparison of bulk vs. Surface challenging.

1) We would like to thank the reviewer for this comment; discussion on the surface/volume NMR signal intensities ratio was missing from the previous version of the article.

Indeed, on the basis of the calculated resonances and the position of the resonating nuclei in the slab, we may calculate the “surface shell thickness” of the various slabs (i.e. the volume where resonating nuclei are influenced by the nearness to the surface). In case of the 0001A slab with thickness 1.85 nm (total of 50 atoms in the super-cell – 17 P), the overall surface thickness is 1.1 nm, covering ~0.594 of the total slab volume (surface layer thickness 0.55 nm). In the case of the $10\bar{1}0$ slab with thickness ~3 nm (total number of atoms in the super-cell 54 – 18 P), the total surface thickness is 1.5 nm, covering ~0.5 of the total slab volume (surface layer thickness 0.75 nm). On the basis of the estimated surface layer thicknesses, and the occupation of each facet, the surface/bulk NMR signal intensities ratios were calculated in all nanoparticle cases **(relevant text has been added in the revised version of the article in page 14 in the case of the 4.3nm nanoparticle, in page 16 in the case of the 6.5nm nanoparticle).**

2) The skin depth for Ni_2P at room temperature and at an NMR frequency of 161.98 MHz is 40 microns, as shown below in the detailed skin depth calculation (Appendix I). This is larger than the usual skin depth of good conductors such as copper, which has a skin depth at this frequency of only 5 microns. The relatively large skin depth of Ni_2P nanoparticles alleviates usual rf inhomogeneity problems encountered in NMR of metallic systems. *For this reason, short high-power adiabatic pulses were used for the acquisition of the bulk Ni_2P spectrum that ensures insensitivity over a much broader variation of rf-field due to the metallic character of the sample than conventional pulses.* This is so, as *the use of adiabatic pulses (DAE pulse sequence), which have the capability to generate uniform rotations of the magnetization even when the rf field (B_1) is highly inhomogeneous¹⁹, compensates for changes in the rf field amplitude, when penetrating the metallic microcrystalline grains of bulk Ni_2P .* The efficiency of this pulse sequence is further confirmed, by comparing the NMR lineshapes of the bulk material with the spectrum of the 40 nm sample, and also with the DFT-calculated NMR spectrum of the bulk material; all spectra are to great extent similar. **The relevant discussion that has been added to address this issue is found in page 21 of the revised main manuscript.**

3) Static NMR lineshapes were acquired with a $\pi/2$ - τ - π Hahn echo. The echo delay 2τ used for the acquisition of the static spectra was set to 60 μ s. **This information has been added to the**

experimental section in page 21 of the revised main manuscript. T_2' vs. Frequency spectra of the microcrystalline (bulk) Ni_2P and the 12 nm nanoparticles have been added in the revised Figure 7. Furthermore, T_2' measurements at peak and intermediate frequencies have been added for the 4.3 nm system in the SI (see Figure 16 of the revised SI manuscript). Two key observations are procured by considering the T_2' data:

- The T_2' vs. Frequency measurements in Figure 7a hold crucial information regarding the encroachment of the P(1) resonances sites towards more negative frequencies, i.e. down to the major peak at around 1700 ppm; this is strikingly similar to the distribution of calculated P(1) frequencies of the $10\bar{1}0$ facet which also advance towards the same frequency range, as shown in Figure 7b. Furthermore, the calculated nuclear magnetization M_0 , obtained from the inversion of the experimental CPMG decays, fits excellently with the DFT-calculated $10\bar{1}0$ spectrum.

- The T_2' values in Figure 7 and SI Figure 16 indicate a minor increase of the relative integral of the signal at 4000 ppm by ~8% (Figures 5c and 6b), which does not affect our conclusions.

The relevant discussion about on this has been added in pages 14 and 19 of the revised version of the article.

(iii)Page 4 - description of the calculated NMR spectra vs. experimental 'an excellent match with the experimental ^{31}P NMR data' - there is a good match between the isotropic shifts in experiment and calculation but the anisotropies as reflected in the spinning sidebands are not reproduced well in the calculations. This should be addressed in the text and the values of the isotropic resonances and shielding parameters (exp and calc) should be provided in the SI.

As implied by the DFT calculations, the orbital term in the Knight shift is responsible for the NMR signal anisotropy. This term is extremely sensitive to the DFT parameterization, i.e. the relaxation scheme used for the optimization of the geometry, which may lead to distortions of the electronic structure of the local Phosphorus environments and therefore, shift anisotropy discrepancies. The sign of the P(2) site shift anisotropy was by error taken opposite. This error has been now corrected in Figure 1b in page 5 of the revised main article. The calculated shift anisotropies are indeed lower than the experimental, as seen in Supplementary Table 1 in page 19 of the revised SI. By increasing only the calculated shift anisotropy values and keeping all other calculation parameters the same for both Ni_{12}P_5 and Ni_2P a match between the calculated and the experimental spectrum is obtained, as shown in the revised Supplementary Figure 12 in page 13. Additionally, we would like to clarify that all anisotropies are following the Haeberlen convention²⁰, which is written as $\Delta\delta = \delta_{zz} - (\delta_{xx} + \delta_{yy})/2$. Additional discussion has now been added in the first paragraph of page 6 of the revised main manuscript.

(iv)Calculated DFT results plotted on top of experimental data in Figure 5-7 with certain linewidth and weight for the distributions. Histograms of the isotropic resonances should be provided for all models. In the description of the fits, the width added to the calculated resonance should be detailed as well as the relative intensity.

Histograms of the isotropic resonances for all models are provided in the revised version of Figure 4 in page 12 of the main manuscript. Linewidths and relative intensities are discussed in the main manuscript in pages 14, 16 and 19 of the revised main manuscript.

(v) Are the nanoparticles capped by ligands? if so can the ligand affect the measured shift? Will ¹H-³¹P CP experiments be helpful in assigning the surface resonances? or the metallic nature of the particle prevents these?

From the ¹H-³¹P CP measurements that have been done to address this comment, no signals that potentially could be attributed to capping ligands or surface resonances could be identified. Among the parameters that were considered was an extremely short contact time to attribute to the short T_2' of the surface signals. Furthermore, FT-IR experiments show no ligand attached to the nanoparticles, exemplified in the revised SI Figure 4b in page 5 for the 12nm nanoparticles.

minor comments:

*figure 13 SI - add legend to the figure. The sequence in the caption does not seem right for T1 measurements, seems like an echo sequence (90-T-180)

*page 3, second paragraph, remove 'induces'

*page 6, second paragraph, line 6, remove 'are'

*page 10, 6th row from bottom: remove 'is the'

These comments have now been addressed.

Appendix I: Skin depth calculation

The Skin depth calculation goes as follows. The distance d that an electromagnetic radiation of frequency ν penetrates into a metal (the classical skin depth) is given by²¹:

$$d = \sqrt{\frac{\rho}{\pi \cdot \mu_0 \cdot \mu_r \cdot \nu}}$$

where ρ is the electrical resistivity of the metal, μ_0 is the vacuum permeability and μ_r is the relative permeability of the metal. The electrical resistivity ρ of Ni₂P at room temperature is $\sim 10^{-4} \Omega\text{cm}^{22}$. No measurements have been reported for the relative magnetic permeability μ_r of Ni₂P. An estimate can be obtained by examining the volume magnetic susceptibility χ since, in the S.I. units (where μ is dimensionless) one has $\chi = \mu_r - 1$. Since Ni₂P is a Pauli paramagnet, its magnetic susceptibility is expected to be very small, in contrast to the striking larger paramagnetic susceptibilities of magnetic ions²³. Therefore, the values should not deviate significantly from 1 as for example happens for all diamagnetic substances. Indeed, the absolute value of the room temperature molar magnetic susceptibility of Ni₂P is $\chi_m = 1.6 \times 10^{-9} \text{m}^3/\text{mol}^{24}$. To convert the molar susceptibility to the dimensionless susceptibility we use the formula $\chi_m = \chi(M/\rho)$ where M is the molar mass and ρ is the mass density. For Ni₂P, $M = 148.394$ and $\rho = 7.33 \text{g/cm}^3$. Substituting we obtain $\chi = 7.90328 \times 10^{-5}$, therefore $\mu_r = 1.00008$. Indeed, the value does not deviate significantly from one. Substituting ρ , and μ_r in the skin depth equation we obtain the value of the skin depth for the Ni₂P at room temperature and for the NMR frequency 161.98 MHz as $d = 39.5 \mu\text{m}$.

References

1. Rundqvist, S. X-Ray Investigations of Mn_3P , Mn_2P and Ni_2P . *Acta Chemica Scandinavica* **16**, 992–998 (1962).
2. Paier, J., *et al.* Screened hybrid density functionals applied to solids. *J. Chem. Phys.* **124**, 154709 (2006).
3. Jain, A. *et al.* Formation enthalpies by mixing GGA and GGA+U calculations. *Phys. Rev. B* **84**, 045115 (2011).
4. Li, J. The electronic, structural and magnetic properties of $\text{La}_{1-1/3}\text{Sr}_{1/3}\text{MnO}_3$ film with oxygen vacancy: a first principles investigation. *Sci. Rep.* **6**, 22422; doi: 10.1038/srep22422 (2016).
5. Meng, Y., *et al.* When Density Functional Approximations Meet Iron Oxides. *J. Chem. Theory Comput.* **12**, 5132-5144 (2016).
6. Pozun, Z. D., and Henkelman G. Hybrid density functional theory band structure engineering in hematite. *J. Chem. Phys.* **134**, 224706 (2011).
7. Janesko, B. G.; Henderson, T. M.; Scuseria, G. E. Screened hybrid density functionals for solid-state chemistry and physics. *Phys. Chem. Chem. Phys.* **11**, 443 (2009).
8. Gao, W., *et al.* On the applicability of hybrid functionals for predicting fundamental properties of metals. *Solid State Communications* **234-235**, 10–13 (2016).
9. Curtarolo, S., *et al.* Accuracy of ab initio methods in predicting the crystal structures of metals: A review of 80 binary alloys. *Calphad* **29**, 163 (2005).
10. Shin, D.; Kim, H.J.; Kim, M.; Shin, D.; Kim, H.; Song, H.; and Choi, S-I.; $\text{Fe}_x\text{Ni}_{2-x}\text{P}$ Alloy Nanocatalysts with Electron-Deficient Phosphorus Enhancing the Hydrogen Evolution Reaction in Acidic Media. *ACS Catal.* 2020, 10, 19, 11665–11673.
11. Bekaert, E., *et al.* Direct Correlation Between the ^{31}P MAS NMR Response and the Electronic Structure of Some Transition Metal Phosphides. *J. Phys. Chem. C* **112**, 20481–20490 (2008).
12. Wexler, R. B.; Martirez, J. M. P.; Rappe, A. M. Stable Phosphorus-Enriched (0001) Surfaces of Nickel Phosphides. *Chem. Mater.* 2016, 28, 5365–5372.
13. Wexler, R. B.; Martirez, J. M. P.; Rappe, A. M. Active Role of Phosphorus in the Hydrogen Evolving Activity of Nickel Phosphide (0001) Surfaces. *ACS Catal.* 2017, 7, 7718–7725.
14. He, J., Morales-Garcia, A., Bludsky, O. *et al.* The Surfaces Stability and Equilibrium Crystal Morphology of Ni_2P Nanoparticles and Nanowires from ab initio Atomistic Thermodynamic Approach. *CrystEngComm* **18**, 3808-3818 (2016).
15. Hansen, M. H., *et al.* Widely available active sites on Ni_2P for electrochemical hydrogen evolution. *Phys. Chem. Chem. Phys.* **17**, 10823 (2015).
16. Li, Q., and Hu, X. First-principles study of Ni_2P (0001) surfaces. *Phys. Rev. B* **74**, 035414 (2006).
17. Nørskov, J. K.; Bligaard, T.; Rossmeisl, J.; Christensen, C. H. Towards the Computational Design of Solid Catalysts. *Nat. Chem.* 2009, 1, 37–46.
18. Sun, S.; Zhou, X.; Cong, B.; Hong, W.; Chen, G. Tailoring the d-Band Centers Endows $(\text{Ni}_x\text{Fe}_{1-x})_2\text{P}$ Nanosheets with Efficient Oxygen Evolution Catalysis. *ACS Catal.* 2020, 10, 9086–9097.
19. Garwood, M. & DelaBarre, L. The return of the frequency sweep: Designing adiabatic pulses for contemporary NMR. *J. Magn. Reson.* **153**, 155–177 (2001).
20. U. Haeberlen, *Advances in Magnetic Resonance*; Suppl. 1; J. S. Waugh, Ed.; Academic Press: New York (1976).

21. J.D. Jackson, Classical Electrodynamics, Second Edition, John Wiley, New York, 1975, p. 298.
22. Shirotani et al, Electrical conductivity of nickel phosphides, Jpn. J. Appl. Phys., Vol. 32 (1993) Suppl. 32-3, pp. 294-296.
23. N.W.Ashcroft and N.D. Mermin, Solid State Physics, Saunders College Publishing, 1976, p. 663.
24. K. Zeppenfeld and W. Jeitschko, Magnetic behaviour of Ni_3P , Ni_2P , NiP_3 and the series $\text{Ln}_2\text{Ni}_{12}\text{P}_7$ ($\text{Ln}=\text{Pr}$, Nd , Sm , Gd-Lu), J. Phys. Chem. Solids, Vol. 54, pp. 1527 - 1531, 1993.

REVIEWERS' COMMENTS

Reviewer #1 (Remarks to the Author):

In this revised manuscript, the authors have supplemented some relevant experiments and discussions to address the raised points. Based on the new results and details, the demonstrations toward the key standpoint become more enriched and rigorous. As a characteristic method, ssNMR could be expected to expand to more nano crystalline materials. Hence, I recommend this work to publish on Nature Communications. However, I intensively suggest the authors further check the crystallographic descriptors in the main text and supplementary information.

Reviewer #2 (Remarks to the Author):

The authors addressed all the revisions requested. The result is a much deeper and careful analysis of their data and interpretation. The results are interesting and should be published, with the minor comment that the new version of the text is a bit too detailed and may be difficult to follow for non experts.

We are grateful to the reviewers and editor for the feedback, which we feel have led to substantial improvement of our manuscript. We have now revised the manuscript according to the points raised in their reports.

Below, please find the original reviewer comments in blue colour and the authors' point-by-point responses in black. All changes outlined in the text below are highlighted in yellow colour in a highlighted version of the revised manuscript.

Reviewer 1

In this revised manuscript, the authors have supplemented some relevant experiments and discussions to address the raised points. Based on the new results and details, the demonstrations toward the key standpoint become more enriched and rigorous. As a characteristic method, ssNMR could be expected to expand to more nano crystalline materials. Hence, I recommend this work to publish on Nature Communications. However, I intensively suggest the authors further check the crystallographic descriptors in the main text and supplementary information.

We are grateful to the reviewer for their positive response to our paper, and for their diligence in the question of the notation for the crystallographic descriptors. These have been carefully checked and corrected in both the main manuscript and SI, according to the standard notation.

Reviewer 2

The authors addressed all the revisions requested. The result is a much deeper and careful analysis of their data and interpretation. The results are interesting and should be published, with the minor comment that the new version of the text is a bit too detailed and may be difficult to follow for non experts.

We are grateful to the reviewer for their positive assessment, and their comment on the readability. To make the manuscript for accessible for non-experts, we have rephrased the following text:

#1

Last Revision: Page 6 line 9 “It is noted that the orbital anisotropy is extremely sensitive to perturbations of the electronic structure of the local environments, which may occur during relaxation of the unit cell and atomic positions compared to the unit cell that is obtained from the Rietveld analysis. This can be observed in the calculated shift anisotropy values of P(1) ($\Delta\delta = -39.35$ ppm) and P(2) ($\Delta\delta = -242.31$ ppm), respectively, which obtain significantly lower values compared to the fitted experimental spectrum, where the shift anisotropy of P(1) is ($\Delta\delta = -123.7$ ppm) and the shift anisotropy of P(2) ($\Delta\delta = -394.2$ ppm). Detailed breakdown of the experimental and calculated parameter values is provided in Supplementary Table 1. Both experimental and calculated shift anisotropies follow the Haeberlen³³ convention, where $\Delta\delta = \delta_{zz} - (\delta_{xx} + \delta_{yy})/2$.”

Replaced by: Page 6 line 10 “(anisotropies are given in the Haeberlen convention³³). Detailed breakdown of the experimental and calculated parameter values is provided in Supplementary Table 1”.

The following text has been added to the caption of Supplementary Table 1 “... with slight deviation in the orbital anisotropy, due to differences between the DFT relaxed unit cell and the unit cell acquired by the Rietveld analysis. Presented experimental and calculated

shift anisotropies are annotated according to the Haeberlen convention, with $\Delta\delta = \delta_{zz} - (\delta_{xx} + \delta_{yy})/2$ and $\Delta\sigma = \sigma_{zz} - (\sigma_{xx} + \sigma_{yy})/2$, respectively. Good match between the experimental and calculated NMR spectra is obtained, by increasing the calculated orbital anisotropy values to the experimental ones, as seen in Supplementary Figure 12”.

#2

Last Revision: Page 14 line 17 “The coherence lifetimes T_2' of the P(1) and P(2) sites measured under static conditions (Supplementary Figure 16) are 244 μs and 448 μs respectively, and so some differential signal loss is expected during the spin echo sequence, which perturbs the measured integrals from their nominal values. Nevertheless, this perturbation is easily calculated and corrected for. Here, both T_2' values are approximately an order of magnitude larger than the half-echo delay of 30 μs in the integral of the resonance at ~ 4200 ppm being $\sim 8\%$ smaller than its nominal value in comparison to the resonance at 1800 ppm. The calculated NMR spectrum was optimized by fitting (i) the intensity ratio of the NMR signals of the (0001-A) and (10 $\bar{1}$ 0) facets, and (ii) the variance of the Gaussian function describing the lineshape (the NMR peaks were simulated by convoluting the calculated shifts with a Gaussian function). To quantify the analysis of the spectrum in two reliable components we allowed the shifts in the calculated spectrum to vary during fitting to match the experimental spectrum and then restore it to the initial value.

Replaced by: Page 14 line 17 “The coherence lifetimes T_2' of the P(1) and P(2) sites measured under static conditions (Supplementary Figure 16) are 244 μs and 448 μs respectively, and so some differential signal loss is expected during the spin echo sequence, which perturbs the measured integrals from their nominal values. Nevertheless, this perturbation is easily calculated and corrected for. Here, both T_2' values are approximately an order of magnitude larger than the half-echo delay of 30 μs , with the result that such effects are small. We calculate that differential dephasing results in the integral of the resonance at ~ 4200 ppm being $\sim 8\%$ smaller than its nominal value in comparison to the resonance at 1800 ppm. The calculated NMR spectrum was optimized by fitting (i) the intensity ratio of the NMR signals of the (0001-A) and (10 $\bar{1}$ 0) facets, and (ii) the variance of the Gaussian function describing the lineshape (the NMR peaks were simulated by convoluting the calculated shifts with a Gaussian function). To quantify the analysis of the spectrum in two reliable components we allowed the shifts in the calculated spectrum to vary during fitting to match the experimental spectrum and then restore it to the initial value.”

Additional changes minor changes

In addition to the changes requested by the reviewers and editor, we have made the following minor revisions:

- In Figure 1, the font type has now been changed uniformly to Arial.
- In Figure 3, the font type has now been changed uniformly to Arial.
- In Figure 4, the font type has now been changed uniformly to Arial.
- In Figure 6, the font type has now been changed uniformly to Arial.
- In Figure 7, the font type has now been changed uniformly to Arial.

- Two new affiliations (3,4) have been added and replaced the old affiliations for the author Kyriaki Polychronopoulou.

- In reference 41 the word Chemrxiv has now been removed.